



# Characterization of an EKO MS-711 spectroradiometer: aerosol retrieval from spectral direct irradiance measurements and corrections of the circumsolar radiation.

Rosa Delia García-Cabrera[1,2], Emilio Cuevas-Agulló[2], África Barreto[3,1,2], Victoria Eugenia Cachorro[1], Mario Pó[4], Ramón Ramos[2], and Kees Hoogendijk[4]

[1]Atmospheric Optics Group, Valladolid University, Valladolid, Spain
[2]Izaña Atmospheric Research Center (IARC), State Meteorological Agency (AEMET), Spain
[3]Cimel Electronique, Paris, France
[4]EKO INSTRUMENTS Europe B.V., The Hague, the Netherlands

**Correspondence:** Emilio Cuevas Agulló
(ecuevasa@aemet.es)

**Abstract.** Spectral direct UV-Visible normal solar irradiance (DNI) measured with an EKO MS-711 spectroradiometer at the Izaña Atmospheric Observatory (IZO, Spain) has been used to determine aerosol optical depth (AOD) at several wavelengths (340, 380, 440, 500, 675 and 870 nm) between April and September 2019 that have been compared with synchronous AOD measurements from a reference Cimel-AERONET (Aerosol RObotic NETwork) sun photometer. The EKO MS-711 has been
calibrated at Izaña Observatory using the Langley-Plot method during the study period. Although this instrument has been designed for spectral solar DNI measurements, and therefore has a field of view (FOV) of 5° that is twice that recommended in solar photometry for AOD determination, the AOD differences compared against the AERONET Cimel reference instrument (FOV ~1.2°), are fairly small. The comparison results between AOD Cimel and EKO MS-711 present a root mean square (RMS) of 0.013 (24.6%) at 340, and 380 nm, and 0.029 (19.5%) for longer wavelengths (440, 500, 675 and 870 nm). However,
under relatively high AOD, near forward aerosol scattering might be significant because of the relatively large circumsolar radiation (CSR) due to the large EKO MS-711 FOV, resulting in a small but significant AOD underestimation in the UV range. The AOD differences decrease considerably when CSR corrections, estimated from LibRadtran radiative transfer model simulations, are performed, obtaining RMS of 0.006 (14.9%) at 340 and 380 nm, and 0.005 (11.1%) for longer wavelengths. The percentage of 2- minute synchronous EKO AOD – Cimel AOD differences within the World Meteorological Organization
(WMO) traceability limits were ≥ 96% at 500 nm, 675 nm and 870 nm with no CSR corrections. After applying the CSR corrections the percentage of AOD differences within the WMO traceability limits increased to > 95% for 380, 440, 500, 675 and 870 nm, while for 340 nm the percentage of AOD differences showed a poorer increase from 67% to a modest 86%.

## 1 Introduction

One of the most important elements that governs the Earth's climate, and its processes, is the presence of atmospheric aerosols.
The most important effect of the presence of aerosols in the atmosphere is the radiative forcing resulting from light scatter-





ing and absorption, and radiation emission. Furthermore, they act as cloud condensation nuclei, modifying cloud properties. Therefore, it is clear that aerosols should be taken into account (IPCC, 2013). Aerosols effect on the Earth Radiation Balance has been quantified as a cooling of -0.45 W m$^{-2}$, and -0.9 W m$^{-2}$ when considering the combined effect of both aerosols and clouds. However, the uncertainty of these values is still very high (WMO, 2016), therefore it is necessary to make more efforts
to evaluate the aerosol atmospheric content and optical properties, such as single scattering albedo, size distribution, etc.

The amount of aerosols present in the atmosphere can be addressed using the aerosol optical depth (AOD) that gives the optical attenuation by aerosols in the atmospheric path. The AOD is derived from surface or satellite observations from sunlight attenuation measurements (WMO, 2016) combined with the Lambert-Beer law. This law has been applied to retrieve the extinction of solar radiation (Ångström, 1930, 1961; Shaw, 1983). The AOD is derived through direct sun radiation mea-
surements at different wavelengths with several instruments such as filter radiometers or spectroradiometers, selecting spectral ranges where the influence of trace gases is minor or even negligible (WMO, 2016; Kazadzis et al., 2018a). The World Meteorological Organization (WMO) recommended the following wavelengths for AOD retrieval: 368, 412, 500, 675, 778 and 862 nm, with a bandwidth of 5 nm (WMO, 1986), and the use of instruments with a full opening angle of 2.5°, and a slope angle of 1° (WMO, 2008).

The AOD retrieval with sunphotometers has been addressed in an extensive list of publications (e.g., Schmid et al. (1999); Kazadzis et al. (2014, 2018a); Barreto et al. (2014); Cuevas et al. (2019)) mainly due to the establishment of aerosol measurement networks, such as AErosol RObotic NETwork (AERONET; Holben et al. (1998)), Precision Filter Radiometer Network (GAW-PFR; Wehrli (2000, 2005)), SKYradiometer NETwork (SKYNET; Takamura and Nakajima (2004)) and SURFace RAdiation Budget Network (SurfRad; Augustine et al. (2008)). Recently, Cuevas et al. (2019) conducted a study comparing AOD
from AERONET-Cimel (1.2° field of view (FOV)) with that from GAW-PFR (2.5° FOV) showing a difference of ∼3 % at 380 nm and ∼2 % at 500 nm compared with AERONET-Cimel for AOD> 0.1, showing GAW-PFR lower values. They demonstrated that this difference was due to the higher amount of dust near-forward scattering measured by GAW-PFR because of it's larger FOV. On the other hand, the AOD retrievals from ground-based spectroradiometers are scarce and normally limited to the visible (VIS) range (e.g. Cachorro et al. (2000); Estellés et al. (2006)). The reason for this shortfall may be found in the
high costs in investment and maintenance of spectroradiometers, and their substantial requirements for calibration compared to sun-photometers. However, spectroradiometers offer the possibility to provide other atmospheric components ($O_3$, $NO_2$, $SO_2$, $CH_4$, $H_2O$, etc) (e.g. Michalsky et al. (1995); Cachorro et al. (1996); Schmid et al. (2001); Barreto et al. (2013); Raptis et al. (2018)).

The first works that attempting to retrieve AOD from spectroradiometers we done by Cachorro et al. (1987) and Ahern et al.
(1991), with results based on a few available data. More recently, several works tackled the AOD multi-spectral retrieval from spectroradiometers with larger datasets. Thus, Cachorro et al. (2000) and Vergaz et al. (2005) reported a quantitative characterization of aerosols in Southern Spain. However, they did not provide a comparison with another AOD retrieval method. Kazadzis et al. (2005) and Gröbner et al. (2001) found AOD differences lower than 0.1 at 355 nm and differences between -0.07 and 0.02 at 315.5 316.75 and 320 nm when comparing AOD retrievals performed with Brewer MKIII and Bentham
DTM 300 spectroradiometers and Li-cor spectroradiometer, respectively. Estellés et al. (2006) retrieved AOD with a Li-cor





spectroradiometers finding differences with Cimel-318 Sun photometers AOD in the 0.01-0.03 (0.02-0.05) range in the VIS range (UV range). Cachorro et al. (2009) compared AOD retrievals from Li-cor and sunphotometer obtaining AOD differences within 0.02 in the range 440-1200 nm. Kazadzis et al. (2018a) presented the results from the fourth WMO filter radiometer comparison for AOD measurements finding an excellent agreement at 500 and 865 nm between PSR (Precision Solar Spec-

troradiometer; Raptis et al. (2018)) and PFR (Precision Filter Radiometer; Wehrli (2008)) and overestimation from 0.01 to 0.03, respectively. López-Solano et al. (2018) compared AOD retrievals from Brewer spectroradiometers, AERONET-Cimel and UVPFR in the range 300-320 nm at Izaña Observatory, with uncertainties lower than 0.05.

In this paper we contribute to the knowledge of spectral AOD with a comparison between AOD from AERONET-Cimel sun photometer (onwards, Cimel AOD) and AOD computed from the direct normal irradiance (DNI) measurements performed

with an EKO MS-711 spectroradiometer (onwards, EKO AOD). We have also addressed the small, but significant, EKO AOD underestimation under relatively high AOD due to dust near-forward scattering, but in this case have compared two instruments whose FOV values show a big difference since the EKO FOV is 5°. We have divided this work into 5 sections: Sect. 2 describes the main characteristics of the Izaña station and the technical description of the instruments used in this work. In Sect. 3 the methodology used to determine AOD and the corrections due to the differences in dust forward scattering, using

the LibRadtran radiative transfer model (RTM) and spectral Langley-Plot calibration are described. In Sect. 4 the main results of the comparison are shown. Finally, a summary and the main conclusions are given in Sect. 5.

## 2 Site description, Instrument and ancillary information

### 2.1 Site description

The data used in this work were acquired between April and September 2019 at the Izaña Observatory (IZO). This observatory

is located on the island of Tenerife (Spain; 28.3°N, 16.5°W; 2.4 km a.s.l.) and it is approximately 350 km away from the African continent. This observatory is managed by Izaña Atmospheric Research Center (IARC) from the State Meteorological Agency of Spain (AEMET) (more information: http://izana.aemet.es; last access: 7 November 2019).

In 1984, IZO enrolled in the WMO Background Atmospheric Pollution Monitoring Network (BAPMoN) and the WMO Global Atmosphere Watch (GAW) program in 1989. IZO collaborates with different international networks such as the Network

for the Detection of Atmospheric Composite Change (NDACC) since 1999, and the GAW-PFR since 2001. In 2003, the Regional Brewer Calibration Centre for Europe (WMO/GAW RBCC-E) was established. Furthermore, IZO has been part of AERONET since 2004, as one of the two AERONET Langley-Plot calibration sites (Toledano et al., 2018). Since 2009, IZO runs a Baseline Surface Radiation Network (BSRN) station. In 2014, IZO was appointed by WMO as a Commission for Instruments and Methods of Observation (CIMO) Testbed for aerosols and water vapor remote sensing instruments (WMO,

2014). More details of IZO programs can be found in Cuevas et al. (2017).





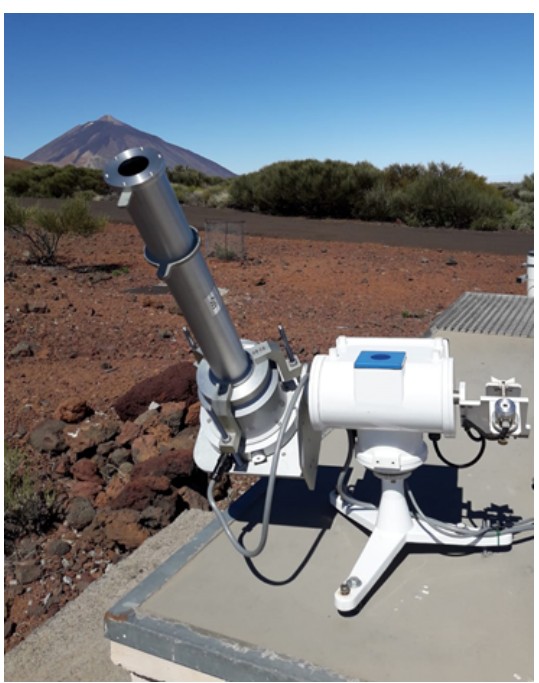

**Figure 1.** The EKO MS-711 spectroradiometer installed at IZO.

## 2.2 Instrument: EKO MS-711 spectroradiometer

An EKO MS-711 grating spectroradiometer used in direct sun measurement mode has been tested (Figure 1) within the CIMO
Testbed program from April to September 2019 (14706 datapoints).

The EKO MS-711 was designed to measure global solar spectral radiation within the 300 and 1100 nm wavelength range
with an average step of ∼0.4 nm, exhibiting a full-width-at-half-maximum (FWHM) <7 nm. It is equipped with its own built-
in entrance optics, and the housing is temperature-stabilized at $25° \pm 5°$ (Egli et al., 2016). EKO Instruments designed a
collimator tube that also allows measuring DNI (see Figure 1).

This spectroradiometer has been mounted on an EKO sun-tracker STR-21G-S2 (accuracy of <0.01°). This setup performs
DNI measurements each 1-minute. The main specifications of the EKO MS-711 spectroradiometer are shown in Table 1.

## 2.3 Ancillary Information: Cimel sun-photometer/AERONET

In this work, we have used AOD data provided by the AERONET permanent Cimel CE318 reference instrument to compare
the AOD derived with the EKO MS-711 spectroradiometer. The different Cimel reference have been shown to have a good
traceability with the world AOD reference (Cuevas et al., 2019). The world AOD reference is maintained by the World Optical
Depth Research and Calibration Center (WORCC) (Kazadzis et al., 2018b).



**Table 1.** Main specifications of the EKO MS-711 spectroradiometer.

| | |
|---|---|
| Wavelength range | 300 to 1100 nm |
| Wavelength interval | 0.3 - 0.5 nm |
| Optical resolution FWHM | < 7nm |
| Wavelength accuracy | ± 0.2 nm |
| Cosine response (zenith: 0-80°) | < 5 % |
| Temp. dependency (-10 °°C to 50 °C) | < 2 % |
| Temp. control | 25 °C ± 2 °C |
| Operating temperature | -10 to 50 °C |
| Exposure time | 10 ms$^{-5}$sec Automatic adjustment |
| Dome material | Synthetic Quartz Glass |
| Communication | RS-422 (Between sensor and power supply) |
| Power requirement | 12VDC, 50VA (from the power supply) |
| Full opening angle (FOV) | 5° |

The Cimel CE318 photometer is an automatic sun-sky scanning filter radiometer that measures AOD at 340, 380, 440, 500, 675, 870 and 1020 nm (nominal wavelength; extended wavelength versions additionally have 1640 nm) with a full opening angle of 1.2°. The uncertainty in AOD measurements from Cimel field instruments, was estimated to be ± 0.01 in the VIS range and near-IR, increasing to ± 0.02 in the UV range (340 and 380 nm) (Eck et al., 1999). This estimate gives an absolute bias > 0.01 for AOD lower than 1.5 (Sinyuk et al., 2012). In this work, we have used cloud-screened and quality-assured
AERONET Version 3.0 Level 1.5 AOD data.

## 3    Methodology

### 3.1    Spectral Langley Calibration

The EKO MS-711 spectroradiometer was factory calibrated by EKO Instruments making use of a calibrated transfer standard 1000 W quartz tungsten-halogen coiled-coil filament lamp that is traceable to the National Institute of Standards and Technol-
ogy (NIST) standard (Yoon et al., 2000). The instrument resultant uncertainty is ± 17% for the UV range, and < 5% for the VIS range. In November 2016, the EKO MS-711 participated in an intercomparison campaign of spectroradiometers at the National Oceanic and Atmospheric Administration (NOAA) Mauna Loa observatory, Hawaii Island, USA (19.54° N, 155.58° W; 3397 m a.s.l.) (Pó et al., 2018), where it was calibrated with the Langley method (Ångström, 1970; Shaw et al., 1973; Shaw, 1983). In 2018 the instrument was deployed at the World Radiation Center- Physical Meteorological Observatory (WRC-PMOD)
for its characterization using a tunable laser (Sengupta et al., 2019). Recently, between April and September 2019, the EKO MS-711 has been calibrated at Izaña Observatory using the Langley method in the 300-1100 nm spectral range. In this study we have used the calibration coefficients with the Langley-Plot method.





The Langley method used in the IZO Langley calibration is based on the Beer-Lambert-Bouguer law:

$$DNI(\lambda) = DNI_o(\lambda)e^{-\tau(\lambda)m} \qquad (1)$$

where $DNI(\lambda)$ is the direct normal irradiance at wavelength ($\lambda$) measured by the instrument, $DNI_o(\lambda)$ is the top-of-atmosphere irradiance corrected for the Sun–Earth distance at wavelength ($\lambda$), $m$ is air mass, and $\tau(\lambda)$ is the optical depth tat can be written in the UV-VIS range as:

$$\tau(\lambda) = \tau_R(\lambda) + \tau_a(\lambda) + \tau_{NO_2}(\lambda) + \tau_{H_2O}(\lambda) + \tau_{O_2}(\lambda) + \tau_{O_3}(\lambda) \qquad (2)$$

where $\tau_R(\lambda)$ is the Rayleigh optical depth due to the molecular scattering that depends on the station pressure as well as on the optical air mass ($m_R$) (Bodhaine et al., 1999), $\tau_a(\lambda)$ is the AOD, and the rest of the terms are the absorption by atmospheric gases in the affected wavelengths (Gueymard, 2001) and are defined as follows:

$$\tau_{NO_2}(\lambda) = u_{NO_2}A_{NO_2} \qquad (3)$$

where $u_{NO_2}$ is the reduced path-length (in atm-cm) taken from the OMI total column $NO_2$ monthly average climatology and $A_{NO_2}$ its spectral absorption coefficient (Rothman et al., 2013).

$$\tau_{H_2O}(\lambda) = (u_{H_2O}A_{H_2O})^b \qquad (4)$$

where $u_{H_2O}$ is the column water vapour content (precipitable water) taken from a Global Navigation Satellite System (GNSS) receiver considering satellite precise orbits at IZO (Romero Campos et al., 2009), $A_{H_2O}$ the spectral absorption coefficient Rothman et al. (2013), and the $b$ exponent depends on the central wavelength position, instrument filter function, as well as the atmosphere pressure and temperature (Halthore et al., 1997). We have determined $\tau_{H_2O}(\lambda)$ from the transmittance for different water vapour and solar zenith angle (SZA) values from the MODTRAN model (Raptis et al., 2018).

$$\tau_{O_2}(\lambda) = (u_{O_2}A_{O_2})^b \qquad (5)$$

where $u_{O_2}$ is the altitude-dependent gaseous scaled path-length taken from the Fourier transform infrared spectrometer (FTIR) measurements at IZO (Schneider et al., 2005), $A_{O_2}$ is the spectral absorption coefficient (Rothman et al., 2013), and the $b$ exponent was obtained from the transmittance values simulated with the MODTRAN model (Berk et al., 2000) for IZO, obtaining a value of 0.454. This value is similar to that obtained by Pierluissi and Tsai (1986, 1987).

$$\tau_{O_3}(\lambda) = (u_{O_3}A_{O_3})^b \qquad (6)$$

where $u_{O_3}$ is the total column ozone obtained with a reference Brewer spectrophotometer at IZO (Redondas et al., 2018), and $A_{O_3}$ the ozone absorption cross section (Brion et al., 1993, 1998).

The Langley-Plot determines $DNI_o(\lambda)$ (that allows to derive calibration constant) from a linear extrapolation of $DNI(\lambda)$ measurements to zero air mass, corrected to mean Sun–Earth distance, and plotted on a logarithmic scale versus air mass:

$$lnDNI(\lambda) = lnDNI_o(\lambda) - [\tau_R(\lambda) + \tau_a(\lambda) + \tau_{NO_2}(\lambda) + \tau_{H_2O}(\lambda) + \tau_{O_2}(\lambda) + \tau_{O_3}(\lambda)]m \qquad (7)$$





**Table 2.** Wavelengths characteristics of Cimel and spectral corrections used in the calculation of AOD.

| Nominal central wavelength (nm) | Filter Bandpass (nm) | Spectral Corrections |
|:---:|:---:|:---:|
| 340 | 2 | Rayleigh, $NO_2$, $O_3$ |
| 380 | 4 | Rayleigh, $NO_2$ |
| 440 | 10 | Rayleigh, $NO_2$ |
| 500 | 10 | Rayleigh, $NO_2$, $O_3$ |
| 675 | 10 | Rayleigh, $O_3$ |
| 870 | 10 | Rayleigh |

## 3.2 AOD retrieval method

The AOD retrievals have been calculated from Eq. 7, as follows:

$$AOD = \frac{1}{m_a}[lnDNI_o(\lambda) - lnDNI(\lambda) - \tau_R m_R - (\tau_{NO_2}(\lambda) + \tau_{H_2O}(\lambda) + \tau_{O_2}(\lambda) + \tau_{O_3}(\lambda))m] \tag{8}$$

150 If we group the gases contributions such as $\tau_{gas}$, the AOD expression is reduced to:

$$AOD = \frac{1}{m_a}[lnDNI_o(\lambda) - lnDNI(\lambda) - \tau_R m_R - \tau_{gas}m] \tag{9}$$

In this work, we have calculated the EKO AOD at the same nominal wavelengths as those of the Cimel (340, 380, 440, 500, 675 and 870 nm) following the methodology used by AERONET (Holben et al. (2001); Giles et al. (2019), and references herein). For this, we have taken into account the spectral corrections shown in Table 2. The 340, 380, 440 and 500 nm wave-155 lengths are corrected from nitrogen dioxide ($NO_2$) absorption, and the optical depth is calculated using the total column $NO_2$ OMI monthly average climatology, and the $NO_2$ absorption coefficient from Burrows et al. (1999).

## 3.3 Corrections in AOD under relatively high CSR

The full opening angle and the FOV are normally used indistinctly in the literature, which should not be confused with the viewing angle. Therefore, we use the term FOV for referring to the full opening angle. As we remarked in the introduction, the 160 WMO recommended for AOD retrieval the use of instruments with FOV lower than 2.5° and slope angle of 1° (WMO, 2008). As the EKO MS-711 was designed for DNI measurements, it has a larger FOV of 5°, twice the WMO recommended value for AOD retrievals. To account for the different geometries, we have applied a correction to the EKO irradiance measurements. In this section we explain the methodology applied to the measurements and comparisons with Cimel AOD.

The DNI measurement implies that a certain amount of diffuse radiation coming from the line-of-sight of the instrument 165 towards the Sun, and an annular region around it, the so-called circumsolar region, is measured together with the DNI coming from the Sun disk ($DNI_{SUN}$). The source of this diffuse radiation, CSR (circumsolar radiation), lies on the scattering processes due to the presence of aerosols and clouds (Blanc et al., 2014) in the atmosphere. This CSR has a high dependence pn particle size the aerosol scattering phase function (Räisänen and Lindfors, 2019) and leads to overestimate DNI. Thus, the





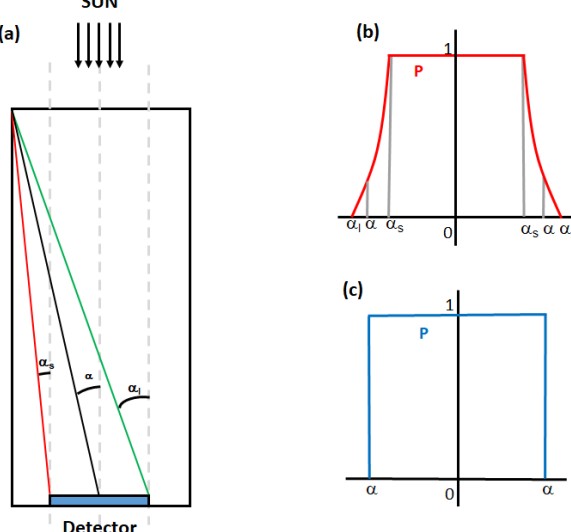

**Figure 2.** (a) Characteristic angles of the instrument: slope angle $\alpha_s$, aperture half-angle $\alpha$ and limit angle $\alpha_l$. On the right, penumbra functions $P(\alpha)$ when (b) the three angles are known and (c) if only the half-angle angle is known. (Figure adapted from Blanc et al. (2014)).

experimental DNI measured by a collimated instrument maybe expressed as the sum of both contributions:

$$DNI = DNI_{SUN} + CSR \tag{10}$$

where $DNI_{SUN}$ is the direct normal irradiance coming from the Sun disk and CSR is the diffuse radiation coming from the sky that is seen by the instrument FOV. This FOV is defined by the instrument geometry and determines the amount of CSR reaching the instrument detector. The value of the DNI measured by the instrument also depends on the atmospheric conditions, and the specific instrument characteristics. The most important element that defines the amount of CSR captured by the instrument is the penumbra function $P$ (Pastiels, 1959) that defines the fraction of Sun radiation captured or not by the collimator, depending on its angle of vision. This penumbra function can be derived from geometrical features of the instrument (Major, 1980; Blanc et al., 2014): the aperture half-angle $\alpha$, the slope angle $\alpha_s$ and the limit angle $\alpha_l$ (Fig. 2a). Usually the three angles are known, being the most important the aperture half-angle $\alpha$. Thus, the radiation coming from the sky with an angle higher than the $\alpha_l$ is outside the collimator and then not measured by the instrument.

If all angles are known the function $P$ takes the shape of Figure 2b, but if $\alpha_s$ and $\alpha_l$ are unknown, the penumbra function $P$ can be approximated as the shape on Figure 2c. In this work, we used the penumbra function $P$ described in Figure 2c, because $\alpha_s$ and $\alpha_l$ are unknown, and considering that $\alpha$ = FOV/2 = 2.5°.





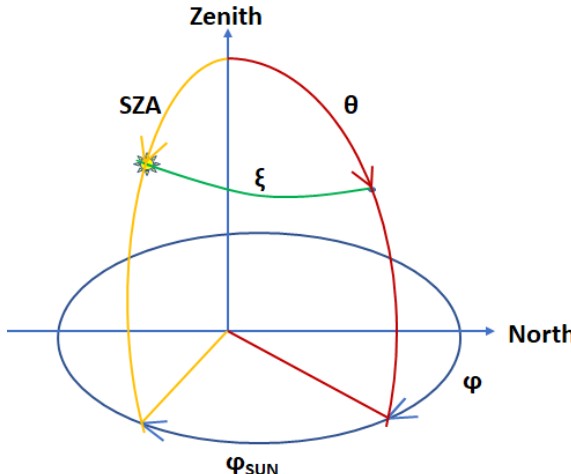

**Figure 3.** Geometry of the problem. The Sun is located in the coordinates (SZA, $\varphi_{SUN}$) and the sky point is in $\theta$, $\phi$. The instrument is located in the origin of the axes.

### 3.4 CSR simulation

Since it is not possible to obtain accurate CSR measurements, it has been simulated with the LibRadtran radiative transfer
model (Mayer and Kylling (2005); Emde et al. (2016), more information http://www.libradtran.org; last access: 7 November 2019), which provides the possibility to simulate the diffuse radiance on sky elements defined by its azimuthal and polar angles. We shortly describe the method followed to simulate the amount of CSR measured by the EKO MS-711. The first step is to describe the geometry of the problem, shown in Figure 3.

For a sky point defined by the polar angle $\theta$ and azimutal angle $\varphi$, the sky radiance on that point is $L\,(\theta,\varphi)$ in W m$^{-2}$sr$^{-1}$.
The angular distance between the considered point and the Sun position (the green arc in Figure 3), is the so-called scattering angle, $\xi$. To obtain the angle $\xi$ of each point on the sky in terms of the polar and azimuthal angles the next equation should be used:

$$cos(\xi) = cos(SZA)cos(\theta) + sin(SZA)sin(\theta)cos(\varphi - \varphi_{SUN}) \tag{11}$$

Taking into account this relation, the radiation field $L$ can be expressed in terms of $\xi$ and $\varphi$, thus the irradiance in the solid
angle subtended by an angular distance from the Sun's centre $\xi$, for an instrument with an aperture half-angle $\alpha$, is (Blanc et al., 2014):

$$I = \int\limits_{0}^{2\pi} \int\limits_{\alpha_o}^{\alpha} P(\xi,\varphi)L(\xi,\varphi)cos(\varphi)sin(\xi) \cdot d\varphi d\xi \tag{12}$$

where $P(\xi,\varphi)$ is the penumbra function defined in Sect. 3.3. If the Sun is in the angular field considered, the obtained irradiance is the DNI of Eq. 10, if not, the result will be only the diffuse radiation. Thus, the key is to simulate the radiances

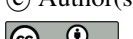



**Table 3.** The inputs to LibRadtran model used in this work.

| Parameters | Input | Reference |
|:---:|:---:|:---:|
| Aerosol parameters | OPAC | Hess et al. (1998) |
| AOD | AOD estimated from EKO MS-711 | - |
| Altitude | 2.4 km | - |
| Absorption Parameterization | REPTRAN (fine resolution) | Gasteiger et al. (2014) |
| Atmosphere profile | Midlatitude summer | Anderson et al. (1986) |
| Solar flux | Kurucz (0.1 nm resolution) | Kurucz (1994) |
| Slit function | Function Gaussian function with FWHM of 6-7 nm | - |
| Radiative transfer equation solver | DISORT, with spherical correction for SZA > 60° | Stamnes et al. (1988) |
| Surface Albedo | 0.11 | García et al. (2014) |
| Ozone Column | Ozone column performed with Brewer spectroradiometer at IZO | - |
| Number of streams | 8 | - |

$L(\xi, \varphi)$ of the points in the FOV that the instrument is "seeing". In this work, and taking into account that the instrument is continuously pointing to the Sun, the integration is performed for $\xi$ values from $\alpha_o = 0.6°$ to $\alpha = 2.5°$ with the aim to simulate the diffuse radiation coming from a circumsolar ring, in order to compare AOD from both instruments using the same CSR.

The input parameters used in the simulations are shown in Table 3. The aerosol contribution has been included in the simulations by using the Optical Properties of Aerosols and Clouds (OPAC package) (Hess et al., 1998). This library provides

the aerosol (and clouds) optical properties in the range 250 nm to 4000 nm. In our case, we focused the interest in the aerosol mixtures, due to the fact the aerosols in the atmosphere are found as a mixture of different particles. In the LibRadtran package are included the aerosol mixtures described in Hess et al. (1998). The aerosol optical properties stored in the datasets used are: the extinction coefficient, scattering coefficient, absorption coefficient, volume phase function, single scattering albedo and asymmetry parameter. Due to the location of the IZO station we have selected the desert mixtures for the cases of low and high

aerosol load respectively.

At this point we should note that the use of 1D simulations with the DISORT (Stamnes et al., 1988) solver implies that the Sun is supposed to be a Dirac function, while, the Sun has an angular radius of $960.12 \pm 0.09"$ (Emilio et al., 2012). However, Stamnes et al. (1988) demonstrated that the error in $DNI_{SUN}$ simulations, when the Sun is assumed to be a point source, is negligible with respect to the finite Sun assumption (Stamnes et al., 2000; Reinhardt, 2013) showed that the simulations

of radiances in the vicinity of the Sun performed using the DISORT and OPAC aerosols for cloud-free cases give the same result than simulations made with the Monte-Carlo RTE solver MYSTIC included in LibRadtran (Mayer, 2009) taking into account the angular extent of solar disk. The differences remain under 1% and even very close to 0%. Since we want to simulate



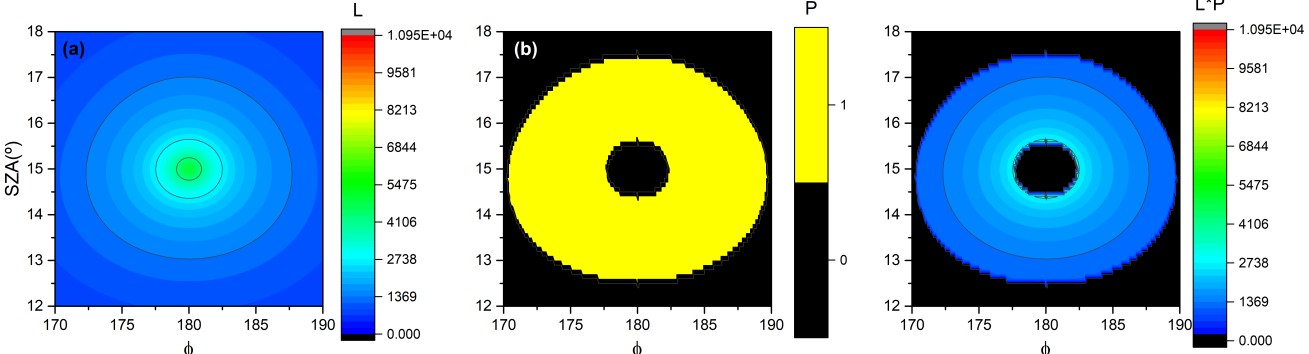

**Figure 4.** Example of the (a) diffuse radiance $L$ (Wm$^{-2}\mu m^{-1}$sr$^{-1}$) at 500 nm shown in colours at different SZA and $\varphi$ (b) penumbra function $P$ determined from Eq 11 and (c) the product of the diffuse radiance $L$ and penumbra function $P$.

cloud-free cases, we can use the 1D, DISORT without introducing significant errors in the simulations against the more precise Monte-Carlo simulations.

Once we have selected the input parameters, we must also select the correct angular grid in azimuthal and polar coordinates to cover, at least, the angular region previously defined ($0.6° \leq \alpha \leq 2.5°$). By using Eq. 11 we can calculate the ranges of polar angles  and azimuthal angles $\varphi$ needed. The result of a monochromatic simulation, i.e. $L(\xi,\varphi)$ at 495 nm for the day 26/07/2019 at SZA of $\sim 14°$ is shown in Figure 4a. In Figure 4b the penumbra function, i.e. $P(\xi,\varphi)$ is shown, and in Figure 4c, the result of multiply $P(\xi,\varphi)\,L(\xi,\varphi)$. Note that the angular grid has been selected in steps of $0.1°$.

The expected CSR will be obtained by integrating the radiation field $P(\xi,\varphi)\,L(\xi,\varphi)$ as indicated in Eq 12. The integration is done by using the angres tool (Mayer and Kylling, 2005) provided in the LibRadtran package which uses a Monte Carlo integration in 2D to obtain the diffuse radiation in the considered radiation field.

### 3.5   AOD retrievals with CSR corrections

Once the CSR has been determined, we apply the correction to the measured DNI taking into account the CSR simulations
explained before. Thus, from Eq. 10 the corrected DNI is:

$$DNI_{CORR} = DNI_{SUN} - CSR \qquad (13)$$

This correction will lead to a $DNI_{CORR} < DNI$, with which we can retrieve an AOD with a similar expression to Eq. 9:

$$AOD_{CORR} = \frac{1}{m_a}[lnDNI_{oCORR}(\lambda) - lnDNI_{CORR}(\lambda) - \tau_R m_R - \tau_{gas}m] \qquad (14)$$

We must note on Eq. 14 that $DNI_o$, calculated with the Langley-Plot calibration method (see Sec. 3.1), should be also cal-
culated applying a FOV correction using Eq. 7, by substituting $DNI_o$ with the corrected $DNI_{oCORR}$. The EKO $AOD_{CORR}$ obtained from Eq. 14 with a $DNI_{oCORR}$ calculated from Eq. 13 is supposed to be "free" of any CSR contribution, then it is straight forward to assume that the $AOD_{CORR}$ is closer to the real AOD present in the atmosphere. In order to know the



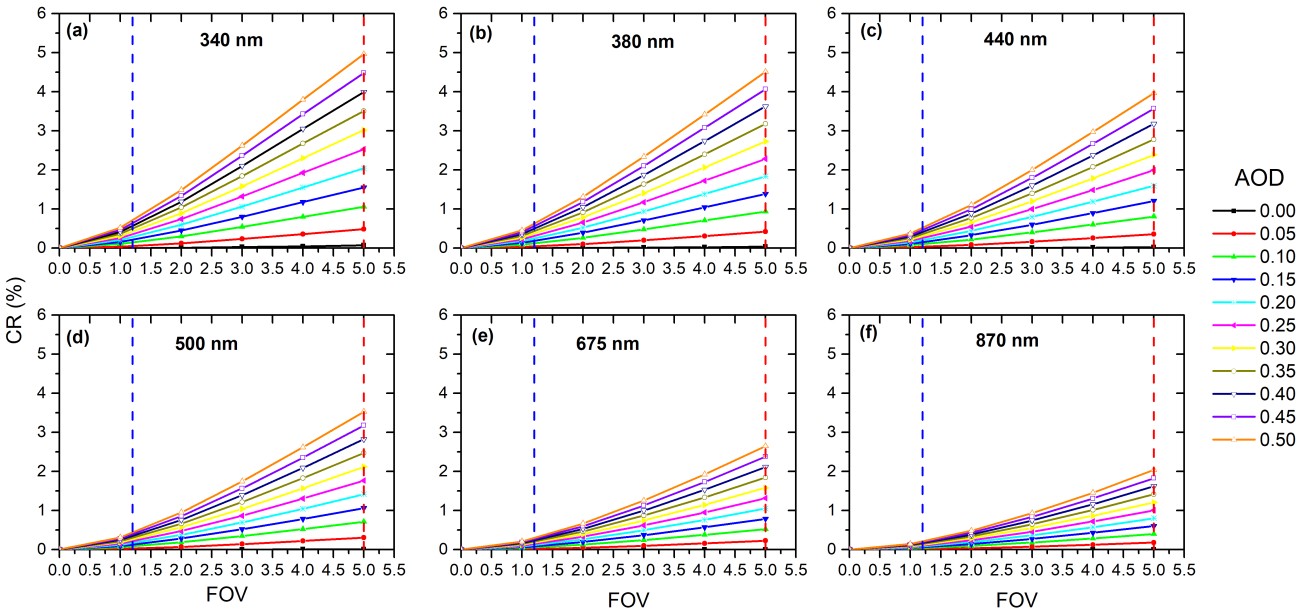

**Figure 5.** Simulations of CR at (a) 340, (b) 380, (c) 440, (d) 500, (e) 675 and (f) 870 nm for AOD between 0.0 and 0.50 and FOV between $0°$ and $5°$ at SZA $30°$. The dashed blue and red lines represent the Cimel FOV ($1.2°$) and EKO MS-711 FOV ($5°$), respectively.

impact of the aerosol load and the FOV size in the values of the CSR simulations we have calculated the ratio of the simulated CSR with respect to the DNI given by Eq. 10, this is the so-called circumsolar ratio (CR) under cloud-free conditions. We have

done simulations of $DNI_{SUN}$ and CSR to obtain the previously cited CR, varying the aerosol load in the range [0-0.50] and the FOV in the range [$0°–5°$]. The rest of the input parameters remain fixed. The results of CR in percentage (Neumann and Witzke, 1999) for a solar zenith angle of $30°$ is shown for the six Cimel channels in Figure 5.

As can be seen in Figure 5, CR increases for higher FOV and larger AOD, as expected, and for the lower wavelengths. The dashed lines in Figure 5 indicate the FOV of the instruments used in this work Cimel (blue line) and EKO (red line). The CR

for the Cimel in all cases is lower than 1% and even 0.5% for the channels over 440 nm. For EKO the CR ranges between 2% in the 870 nm channel and 5% for the 340 nm channel. Thus, the expected CSR maximum values in Figure 5 should be found at these conditions: FOV of $5°$, AOD of 0.50 and wavelength of 340 nm, in which a CR of 5% is found. We have simulated the AOD retrievals as a function of CSR. By combining Eq. 13 to 15, we can vary CR (in fact the value of CSR) and calculate the AOD retrieved with the corresponding $DNI_{oCORR}$.

These results indicate that the CSR impacts significantly on the EKO AOD retrievals under relatively high AOD leading to AOD underestimation, with this effect being less important for the Cimel AOD retrievals because of its narrower FOV.





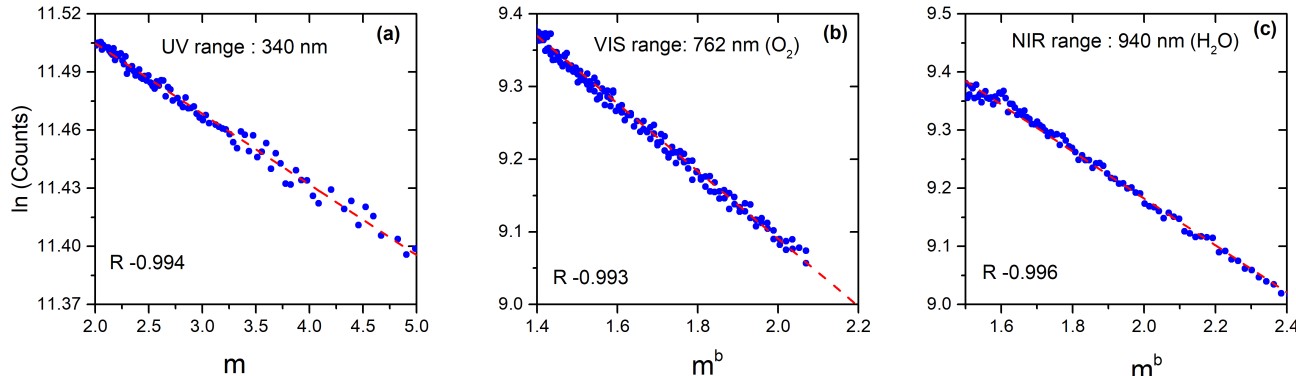

**Figure 6.** Examples of Langley-Plots using the UV-VIS-near IR direct-Sun measurements on 19 March 2019 at Izaña Observatory at (a) 340 nm, (b) 762 nm (O2) and (c) 940 nm (H2O) nm. R: correlation coefficient.

## 4   Results

### 4.1   Langley calibration at Izaña Observatory

Based on the experience of Kiedron and Michalsky (2016) and Toledano et al. (2018), we have considered that the Langley
calibration is suitable if the standard deviation ($\sigma$) of the fit (Eq. 7) is lower than 0.006, the correlation coefficient (R) > -0.99,
the number of valid points > 33% of the initial sample, and AOD (500 nm) < 0.025. In order to test the Langley method
described in Sect. 3.1, an example of Langley-Plots using the UV-VIS-near IR direct-Sun measurements on 19 March 2019 at
Izaña Observatory are shown in Figure 6.

The comparison between the factory calibration performed by EKO Instruments in 2016 and the IZO Langley-Plot calibra-
tion (2019) is shown in Figure 7. This results indicate a high stability of EKO MS-711, in the range 300 nm – 1100 nm, in the
last 3 years. The factory calibration and the IZO Langley-Plot calibration three years later present differences $\sim 4.8\%$ between
350 and 1100 nm and even $\leq 2.3\%$ and $3.1\%$ in the VIS and near-IR range, respectively. The larger differences below 350
nm are attributed to the low halogen lamp signal in this region experienced during the factory calibration, and low instrument
sensitivity in this region.

265    Applying the previous method, $DNI_o(\lambda)$ values and their standard deviations from the EKO MS-711 measurements (from
April to September 2019 at Izaña Observatory) at the nominal wavelengths measured by the Cimel (340, 380, 440, 500, 675
and 870 nm), as a function of time are shown in Figure 8. These $DNI_o(\lambda)$ values have been used in the AOD retrievals.

### 4.1.1   AOD retrievals

In this section, we present the results obtained when comparing Cimel AOD and EKO AOD with no CSR corrections (CSR un-
270    corrected AOD) and applying a CSR correction (CSR corrected AOD). The comparisons were done considering measurements





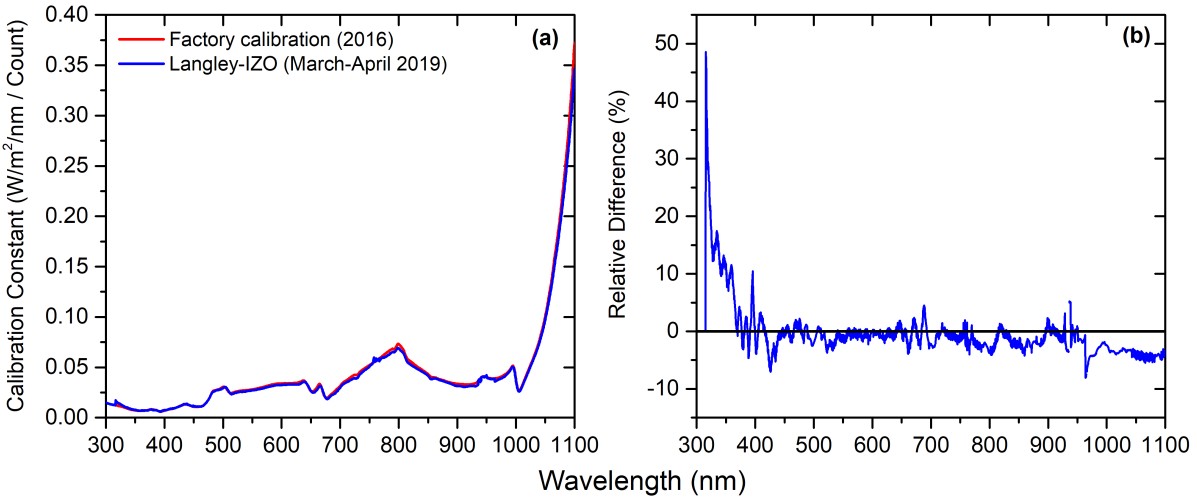

**Figure 7.** (a) Calibration constants (W m$^{-2}$/nm/Count) of the EKO MS-711 spectroradiometer, and (b) relative differences between factory calibration constants and those obtained from Langley-Plots at at IZO.

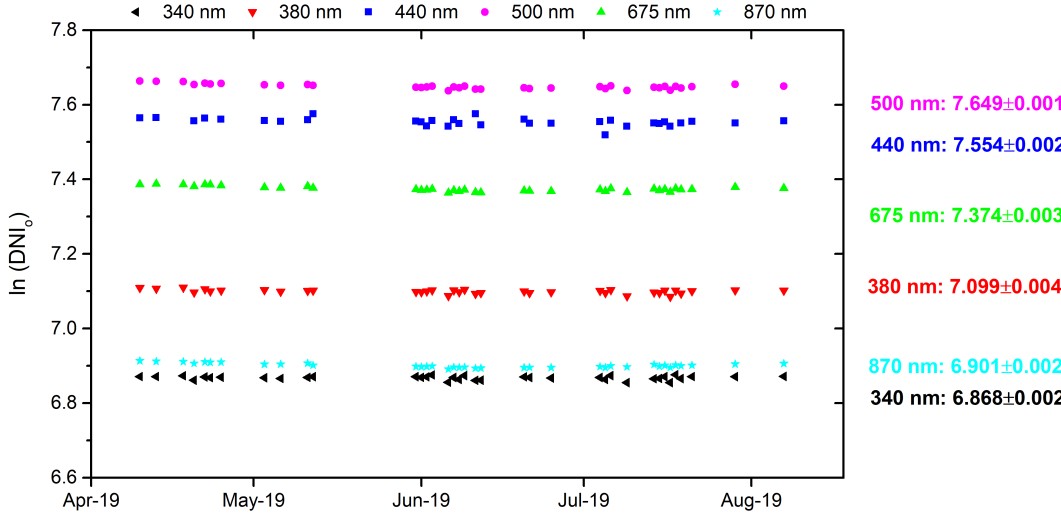

**Figure 8.** EKO MS-711 $DNI_o(\lambda)$ values, and corresponding standard deviations, between April and September 2019 at IZO, for all nominal wavelengths measured by the Cimel (340, 380, 440, 500, 675 and 870 nm).



**Table 4.** Statistics of the comparison between EKO AOD, with no CSR corrections and implementing CSR corrections, and Cimel AOD at 340, 380, 440, 500, 675 and 870 nm at IZO between April and September 2019. R: correlation coefficient, slope of the least-squares fit between EKO AOD and Cimel AOD, RMS: root mean square of the bias and MB: mean bias. The results of the relative bias are in brackets (in %).

| Wavelength | R | | Slope | | RMS | | MB | |
|:---:|:---:|:---:|:---:|:---:|:---:|:---:|:---:|:---:|
| (nm) | CSR | CSR | CSR | CSR | CSR | CSR | CSR | CSR |
| | Unc. | Unc. | Unc. | Unc. | Unc. | Unc. | Unc. | Unc. |
| **340 nm** | 0.960 | 0.973 | 1.063 | 0.994 | 0.017 | 0.007 | 0.015 | <0.001 |
| | | | | | (28.9%) | (16.9%) | (24.5%) | (-1.4%) |
| **380 nm** | 0.981 | 0.986 | 1.071 | 1.001 | 0.009 | 0.005 | 0.007 | <0.001 |
| | | | | | (20.2%) | (12.9%) | (14.8%) | (1.2%) |
| **UV-Range** | 0.971 | 0.979 | 1.067 | 0.997 | 0.013 | 0.006 | 0.011 | <0.001 |
| **(Mean)** | | | | | (24.6%) | (14.9%) | (19.7%) | (1.3%) |
| **440 nm** | 0.984 | 0.987 | 1.041 | 0.997 | 0.101 | 0.005 | 0.009 | 0.001 |
| | | | | | (22.4%) | (13.5%) | (18.7%) | (0.6%) |
| **500 nm** | 0.988 | 0.991 | 1.075 | 1.018 | 0.007 | 0.005 | 0.004 | 0.002 |
| | | | | | (18.2%) | (12.9%) | (12.1%) | (0.4%) |
| **675 nm** | 0.989 | 0.991 | 1.057 | 1.013 | 0.006 | 0.006 | 0.003 | <0.001 |
| | | | | | (19.7%) | (10.7%) | (11.2%) | (0.5%) |
| **870 nm** | 0.998 | 0.999 | 1.039 | 1.009 | 0.004 | 0.003 | <0.001 | <0.001 |
| | | | | | (18.8%) | (7.3%) | (0.3%) | (0.2%) |
| **VIS-Range** | 0.989 | 0.992 | 1.053 | 1.009 | 0.029 | 0.005 | 0.004 | <0.001 |
| **(Mean)** | | | | | (19.5%) | (11.1%) | (10.6%) | (0.4%) |

of both instruments that match within 2 minutes for all wavelengths. This approach produced a Cimel and EKO AOD dataset with a total of 14706 quasi-coincident measurements. The results show (Table 4) that there is a good agreement between EKO AOD and Cimel AOD for all channels, even for no CSR correction, except for the lowest 340 nm UV channel.

The uncorrected EKO AOD shows slopes ~1.06 and correlation coefficients over 0.97 for all wavelengths. The RMS ranges from 0.017 (28.9%) at 340 nm to 0.004 (18.8%) at 870 nm. These results improve significantly when taking into account the CSR corrections for all wavelengths. Thus, for the corrected EKO AOD the correlation coefficients are ~0.98 for the shorter wavelengths and ~1 for the rest of the wavelengths. The RMS and MB show the same trend as that for the uncorrected EKO AOD case, that is, we find the lowest values for the higher wavelengths. The negative values of the MB (EKO AOD – Cimel AOD), indicate that the EKO AOD values are normally lower than the Cimel AOD values. However, these values are within the Cimel instrument uncertainties, ±0.01 in the VIS and near-IR and ±0.02 in the UV ranges (Eck et al., 1999). These results also agree with other studies. For example, Estellés et al. (2006) and Cachorro et al. (2009) found differences between 0.01 and 0.03 in the VIS range, and between 0.02 and 0.05 for the UV when comparing Li-cor AOD with Cimel AOD. Recently, Kazadzis



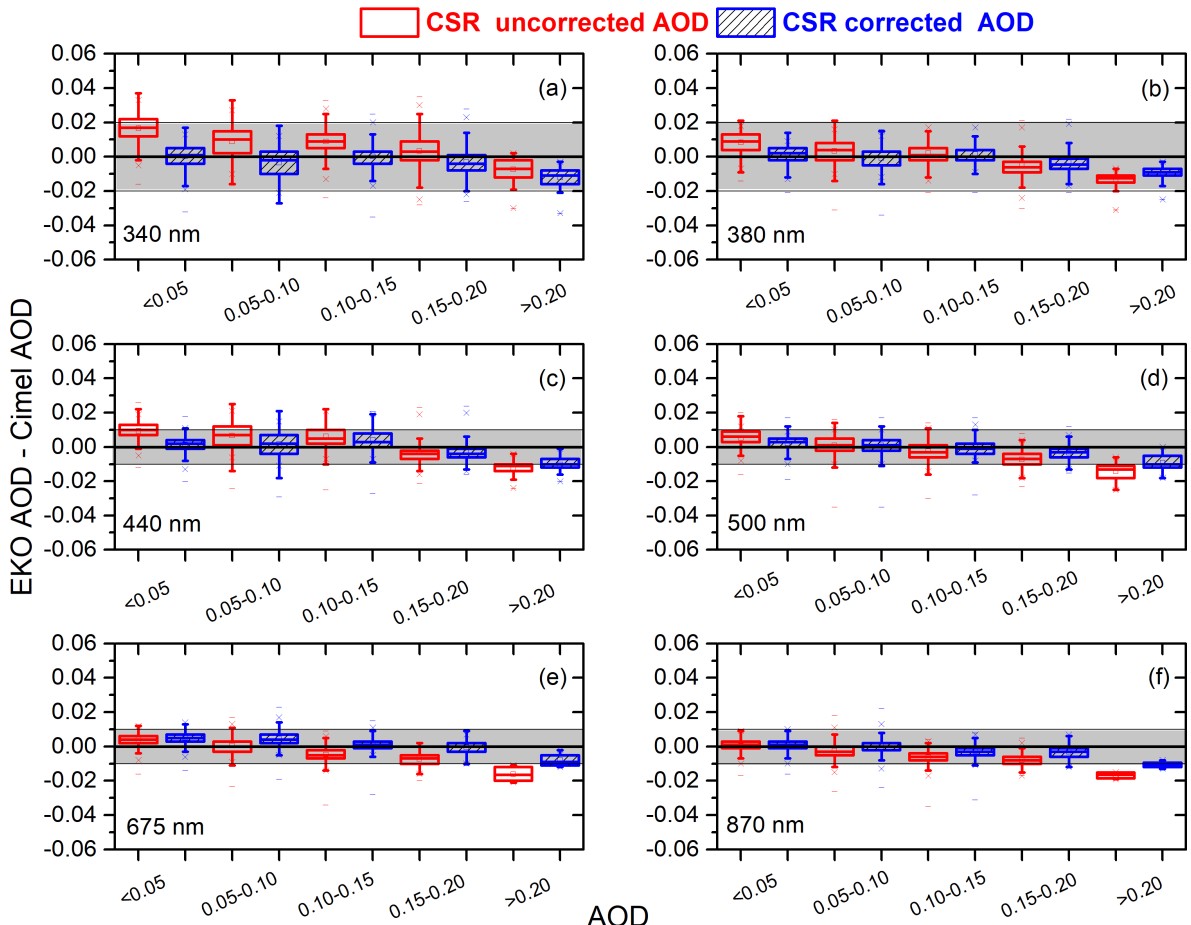

**Figure 9.** Box plot of the differences between the EKO AOD with (no) CSR corrections, and Cimel AOD versus AOD for the period April-September 2019 at IZO in blue (red). Lower and upper boundaries for each box are the $25^{th}$ and $75^{th}$ percentiles; the solid line is the median value; the crosses indicate values out of the 1.5-fold box area (outliers); and hyphens are the maximum and minimum values. Shadings show the range of uncertainty of Cimel ($\pm0.02$ for the UV range and $\pm0.01$ for VIS and near-IR ranges; Eck et al. (1999)).

et al. (2018a) found AOD differences ranging between 0.01 and 0.03 at 500 and 865 nm, respectively, when comparing AOD from PSR and PFR. Recently, Cuevas et al. (2019), using long-term AOD data series from both GAW-PFR and AERONET-Cimel radiometers reported differences in AOD $\sim$3% lower at 380 nm and $\sim$2% lower at 500 nm for GAW-PFR due to its larger FOV.

The box plots of MB differences (EKO AOD – Cimel AOD) for different AOD intervals are presented in Figure 9. In general, it can be seen that a significant improvement in the AOD retrievals is found after CSR correction, with the corrected AOD medians being closer to 0 in all wavelengths. The improvement in AOD for AOD>0.1 conditions is also remarkable mentioned as already in the CSR correction section. The scatter also is significantly reduced for all wavelengths and aerosol loads, except





**Table 5.** Linear AOD-correction equations (slope and intercept) at 340, 380, 440, 500, 675 and 870 obtained with data measured from April 1st to July 31$^{st}$ 2019 at Izaña Observatory. Validation of the linear AOD-correction equations was performed using data obtained between August 1$^{st}$ and September 30$^{th}$ 2019.

| | **Linear AOD-correction Equations:** | | | **Validation** | | |
| | **Corrected EKO AOD = Slope*EKO AOD + Intercept** | | | **01/08/2019-30/09/2019** | | |
| | **01/04/2019-31/07/2019** | | | | | |
| **Wavelength (nm)** | **Slope** | **Intercept** | **R** | **RMS** | **MB** | **R** |
| **340** | 1.076 | -0.019 | 0.997 | 0.005 (5.9%) | -0.003 (-4.0%) | 0.998 |
| **380** | 1.073 | -0.0102 | 0.999 | 0.003 (2.9%) | -0.003 (-1.6%) | 0.999 |
| **440** | 1.066 | <0.001 | 0.999 | 0.002 (2.4%) | <0.001 (-1.2%) | 0.999 |
| **500** | 1.056 | -0.005 | 0.999 | 0.002 (2.9%) | -0.001 (-2.1%) | 0.999 |
| **675** | 1.043 | 0.003 | 0.999 | 0.001 (2.4%) | <0.001 (-1.7%) | 0.999 |
| **870** | 1.031 | <0.001 | 0.999 | <0.001 (1.4%) | <0.001 (-0.02%) | 0.999 |

in the 340 nm UV channel, attributed to the higher instrument uncertainty, as well as a poorer model aerosol characterization this range (see Sect. 2.2). Since the 340 nm and 380 nm channels have 2 nm and 4 nm bandpass, respectively, and the EKO MS-711 FWHM is ∼ 7nm (Table 1), these two UV channels have some additional radiation contribution from the adjacent wavelengths, increasing their uncertainty and causing an AOD overestimation. Despite these drawbacks, the improvement in AOD is significant performing a simple correction of the CSR estimated with LibRadtran.

The linear AOD-correction equations were determined by using data measured from April 1$^{st}$ to July 31$^{th}$ 2019 at Izaña Observatory (Table 5). The validation of these linear AOD-correction equations was performed using an independent period of data (between August 1st and September 30$^{th}$ 2019). Note that MB ≥-1.6% for all wavelengths except for 340 nm for which a significantly larger MB (-4.0%) is registered. In any case it should be noted that the CSR correction applied in this study has been made under the presence of mineral dust. It would be necessary to verify that these CRS corrections have similar validity under moderate-high influence of other types of aerosols, such as marine or biomass burning aerosols.

In order to check the quality of EKO AOD, we have applied the WMO traceability criteria (WMO, 2005) defined for finite FOV instruments as:

$$U_{95} = \pm(0.005 + 0.010m_a) \tag{15}$$

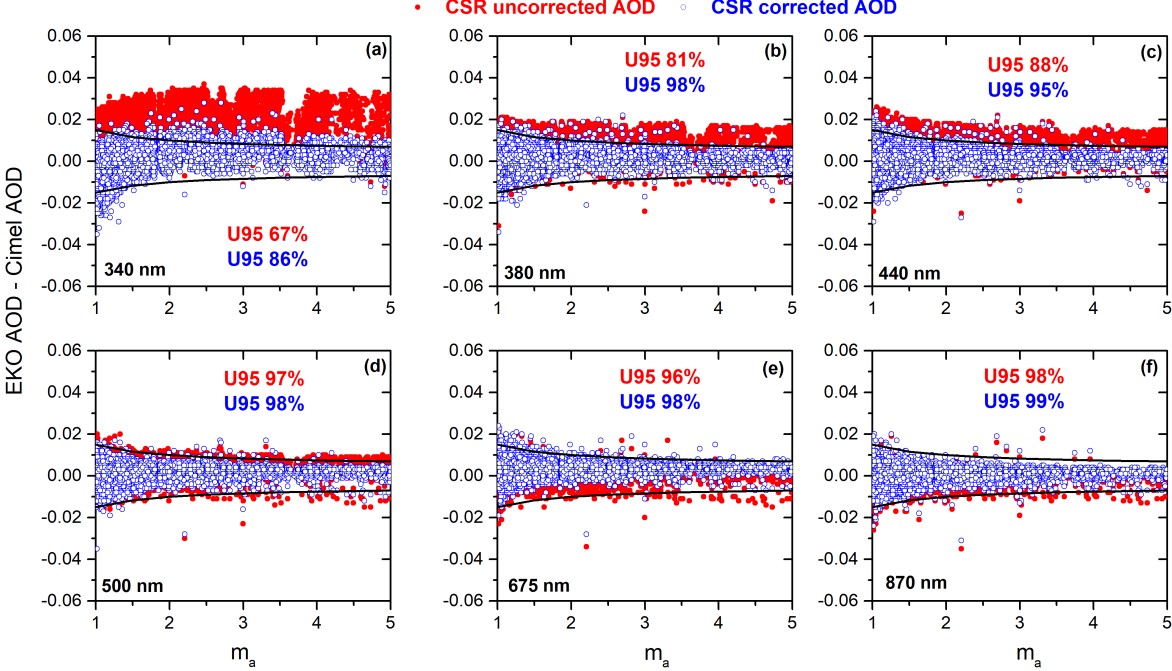

**Figure 10.** AOD differences (EKO AOD – Cimel AOD) versus the optical air mass ($m_a$). Black lines represent the $U_{95}$ uncertainty limits.

where $m_a$ is the optical air mass. The percentage of data meeting the WMO traceability requirements (95% of the AOD differences of an instrument compared to the WMO standards lie within specific limits) is > 95% at 500, 675 and 870 nm, taking the AERONET-Cimel as the reference (Figure 10).

The percentage of EKO AOD data meeting the WMO criteria increases considerably when we take into account the CSR corrections, increasing from 67% to more than 86% at 340 nm, and above 95% for the rest of the channels. The poorest results shown by the 340 nm channel (86%), might be partially explained by the. EKO's 7nm FWHM influence on the smaller 2 and 4nm band pass UV channels. The instrument uncertainty is larger in the UV range, which is mostly associated with stray-light in the instrument inner optics (Zong et al., 2006).

When focusing the analysis on relatively high AOD (AOD> 0.10), we found that the percentage of AOD differences out of the WMO $U_{95}$ limits were ~3.5% at 380 nm and 0.6% at 870 nm, consistent with the lower percentages of AOD differences out of the WMO $U_{95}$ reported by Cuevas et al. (2019) when comparing GAW-PFR (FOV ~ 2.5°) and AERONET-Cimel radiometers that present a lower difference in FOV (1.2°).





# 5 Conclusions

In this work, we present the characterization of an EKO MS-711 spectroradiometer. The instrument has been calibrated at Izaña Observatory using the Langley-Plot method between April and September 2019. This calibration has been compared
with the lamp calibration performed at EKO Instruments factory in 2016, obtaining relative differences ≤2.3% and 3.1% in the VIS and near IR range, respectively. These results indicate a high spectral stability of the instrument in this 3-year time period (2016-2019).

The EKO MS-711 has been designed for spectral solar DNI measurements, and therefore it has a relatively high FOV (5°), double the FOV recommended by WMO for AOD radiometers, and four times larger than the AERONET-Cimel FOV.
This difference in FOV might lead to a significant difference in near forward scattering under relatively high aerosol content, resulting in a small, but significant, AOD underestimation, especially in the UV range.

However, the AOD retrievals from an EKO MS-711 spectral DNI measurements show a rather good agreement with those from an AERONET reference radiometer. The AOD comparison was held at Izaña Observatory between April and September 2019. Quality assessment of the EKO MS-711 AOD has been performed by comparing with coincident AOD from AERONET
at 340, 380, 440, 500, 675 and 870 nm considering measurements of both instruments as close as 2 minutes between them, with a total of 14706 analyzed data-pairs. The skill scores of the AOD comparison are fairly good with a RMS of 0.013 (24.6%) at 340 and 380 nm, and 0.029 (19.5%) for longer wavelengths (440, 500, 675 and 870 nm), with AOD being underestimated by the EKO radiometer. The MB (EKO AOD – Cimel AOD) are 0.011 (19.7%) for 340 and 380 nm and 0.004 (10.6%) for 440, 500, 675 and 870 nm. These results improve considerably when we take into account the CSR corrections to EKO AOD
because of the higher EKO FOV. The CSR differences between EKO and AERONET-Cimel were obtained using LibRadtran model. When comparing EKO AOD corrected values the RMS is reduced to 0.006 (14.9%) at 340 and 380 nm, and to 0.005 (11.1%) for longer wavelengths, while MB is reduced to <0.001 (1.3%) for 340 and 380 nm, and <0.001 (0.4%) for 500, 675 and 870 nm. These values are within the Cimel instrumental uncertainty (±0.01 in the VIS and near-IR, and ±0.02 in the UV ranges).

Following WMO recommendations we have analysed the percentage of EKO AOD – Cimel AOD differences within WMO $U_{95}$ limits defined for finite FOV instruments, we found that with no CSR-corrections ≥96% of the AOD differences fell within the WMO $U_{95}$ limits at 500, 675 and 870 nm. After applying the CSR-corrections, the percentage of AOD differences within the WMO $U_{95}$ limits were >95% for 380, 440, 500, 675 and 870 nm, while for 340 nm the percentage of AOD differences within the WMO $U_{95}$ increased only to a modest 86%. The known greater AOD uncertainty in the UV range along with stray-
light problems not fully corrected in this instrument seem to be behind the poorer AOD agreement with AERONET-Cimel at 340 nm.

The EKO-MS711 has proven to be an instrument, which despite having been designed for solar radiation measurements, can provide high quality AOD measurements in the VIS and near-IR ranges with excellent results when compared with the AERONET-Cimel reference radiometer, which, in turn has shown a very good AOD traceability with the WORCC AOD world
reference.



*Data availability.* The Cimel-AERONET data from the Izaña station ("Izana") are available from the AERONET website: https://aeronet.gsfc.nasa.gov/ (last access: 14 November 2019). The EKO MS-711 data might be available upon request to EKO Instruments and Izana WMO CIMO Testbed.

*Author contributions.* RDG and EC designed the structure and methodology of the paper and wrote the main part of the manuscript. RDG
computed all the calculations performed in the paper. AB discussed the modelling results and participated in the AOD retrieval and the Langley calibration analysis. VEC provided interesting ideas used in this paper, and advice based on her experience in spectroradiometry. RR performed the maintenance and daily checks of the EKO-MS711 spectroradiometer. MP provided detailed technical information and calibrations of the EKO-MS711 spectroradiometer. KH allowed that the EKO-MS711 used in this study could be evaluated in the WMO CIMO Izaña testbed taking care of all the associated logistics. All authors discussed the results and contributed to the final paper.

*Competing interests.* The authors declare that they have no conflict of interest.

*Acknowledgements.* This work has been developed within the framework of the activities of the World Meteorological Organization (WMO) Commission for Instruments and Methods of Observation (CIMO) Izaña test bed for aerosols and water vapor remote sensing instruments. The authors are grateful to EKO Instruments for its availability that the EKO-MS711 spectroradiometer has been tested and evaluated independently by the WMO CIMO Izaña testbed. The LibRadtran Radiative Transfer Model has been used to estimate the circumsolar
radiation. AERONET Sun photometers at Izaña have been calibrated within the AERONET Europe TNA, supported by the European Union Horizon 2020 research and innovation program under grant agreement no. 654109 (ACTRIS-2). This research benefited from the results of the project funding by MINECO RTI2018-097864-B-I00. We also acknowledge our colleague Celia Milford for improving the English of the manuscript.



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
