# Peer review of "Aerosol retrievals from the EKO MS-711 spectral direct irradiance measurements and corrections of the circumsolar radiation."

_Atmospheric Measurement Techniques, 2019_

## Referee Comment (RC1) · Lionel Doppler (Referee) · 14 Jan 2020

Reviewer Comments to Garcia et al., AMTD 2017

From: Lionel Doppler, Deutscher Wetterdienst / Meteorologisches Observatorium Lindenberg – Richard Assmann Observatorium (DWD/MOL-RAO), Lindenberg (Tauche), Germany.

• Copernicus article references:

[Figure]

Journal: AMT

Title: Characterization of an EKO MS-711 spectroradiometer: aerosol retrieval from spectral direct irradiance measurements and corrections of the circumsolar radiation

Author(s): Rosa Delia García-Cabrera et al.

MS No.: /amt-2019-467

MS Type: Research article

1. General Comments justifying the evaluation

This paper presents a method to retrieve aerosol optical depth (AOD) out of spectral DNI (direct sun normal irradiance) radiation measurements from the spectroradiometer EKO MS-711. The paper presents the instrument, the site of the observations (IZO: Izaña Atmospheric Observatory), and the method used. An issue that is well discussed is how to correct the measured DNI, obtained with the EKO instruments that has a larger field of view than the WMO standards suggest for AOD measurements. The solution found is to estimate the CSR (circumsolar radiation) by simulating the forwarded scattered radiation with a radiative transfer code and multiplying it with a so-called penumbra function depending on the solar angles (azimuth and zenith). The method of AOD inversion is validated thanks to a comparison to a reference instrument (the Cimel – Aeronet photometer) for six wavelengths in UVA, VIS and NIR at the site of IZO during four months (April – July 2019). A statistical study is presented to validate the AOD retrieval method and evaluate the gains of the CSR correction.

The most innovative part of the paper is the presentation of the CSR estimation and the correction of the DNI for this instrument having a field of view of 5° in order to be compared to photometers having a field of view of less than 1.2° (WMO standards). This method is well explained in the paper and the reader can be convinced of the reliability of it.

The main concept presented in the paper is the AOD retrieval out of spectral DNI measurements from a spectroradiometer, this is not new, but only few articles are making a detailed presentation of the method explaining each step and showing all the equations. This is well done in this paper and will be useful for the AOD community, the photometer community and the spectroradiometer community.

The validation of the method is shown thanks to a detailed statistic comparison to a reference instruments, mentioning WMO traceability criteria and discussing objectively, fairly and humbly the weak points of the method and instrument. Thus, substantial conclusions are reached: the paper evaluates quantitatively the DNI correction method, the AOD retrieval method and its application to the instrument EKO MS-711, convincing the readers that these methods can be used operationally with this instrument.

The scientific methods used are well described their validity are discussed, a good balanced use of figures and mathematic equations contributes to a clear outline of them.

The references list is complete enough giving proper credit to current and past work related to this topic. The number of references is good balanced and the references are of excellent quality. Thanks to this literature work, the authors could clearly put forward their own contribution to the topics approached in this paper.

The title of the paper reflects the content of the paper in a good way; the abstract is a good complement of the title and a concise and truth summary of the paper.

The overall presentation is well structured, and despite some minor details (to which I suggested improvements in the part below named "technical comments") clear expressed. The language is fluent and precise and it is an obstacle neither to get rapidly a good comprehensive view of this work nor to understand the technical and mathematical details The mathematical formulae are shown in a good way. The equations are correct written, without mistake and well understandable.

I would suggest some minor improvements to be done: A table with all acronyms would

be welcome. Also I join a list of technical corrections (see below: "technical comments"). Moreover some points should be briefly discussed, these questions are asked below in "specific comments". These are minor/technical corrections that I suggest.

Despite these technical corrections that have do be done, the article is of good scientific quality, of good significance and of good presentation. This justifies my evaluation here above and the fact that I suggest the editor to accept the manuscript and to ask for technical corrections and to answer to the four questions mentionned here below in "specific comments / questions"

2. Specific Comments / questions

- About the CSR correction presented in 3.4.: The simulated forwarded scattering radiation is computed using desert dust aerosol. How can we adapt the correction factors to other type of aerosols? And if it is possible: How is it possible to integrated the characterization of the aerosol kind in an operational algorithm in order to have directly the CSR correction factors suitable to the defined aerosol type?

- IZO is a site of low aerosol amount. The results presented in the statistical study to validate the method (part 4.) shows AOD ranging between 0.0 And 0.2 (eg: Figure 9). How many points of comparison do you have for AOD > 0.15? What do you expect it should happen for other sites having larger AOD (continental sites in middle Europe or close-urban areas)?

- Are the results shown in Part 4 restricted to cases with desert dust aerosols? If yes, do you have some preliminary results for other kinds of aerosols? What do you expect it should happens? If no (= the results shown corresponds to different mixtures and kinds of aerosols), do you have some differences between different kinds of aerosols detected?

- The AOD retrieval method presented in 3.1 and 3.2 is well described. Nevertheless, I would discuss two points more in detail: 1) Do you take the same airmass for

aerosols, water vapour, mixed gases and ozone? 2) How do you compute Rayleigh optical depth? With which formula (Bodhaine ?) and with which values of the air pressure (Aeroent uses a 6 hours average taken from a model)?

3. Technical comments

General comment: Please introduce a list of all acronyms used

Abstract:

- At the beginning of the abstract, should be explained what is the spectral range and resolution of EKO MS-711.

Introduction

- L25: "properties, such as single scattering albedo, size distribution, etc" -> please avoid "etc", write a complete list, best sorted in decreasing importance order.

- L47: (again etc.) -> please complete list or use "e.g.:"

- L47: reference is Barreto 2014 (and not 2013) for spectroradiometer and Aerosol.

- L67: Go to next line before presenting the parts of your papers with "We have devided..."

Part 2: Site Description, Instrument and ancillary information

2.2 Instrument: Maybe explain what kind of technology it is: monochromator or array spectrometer (it is not specified).

- L98: Specify in this part of the text that the world AOD reference is the PFR in order that the reader knows from which instrument you are talking about.

- L104: Bias < 0.01 (and not > 0.01) (citing Sinyuk, GRL 2012)

Part 3. Methodology

- L130+L136+L141, maybe use a different description for "b" of each gas: b_H2O,

b_O2, b_O3 for example. Here you have the same letter and the reader can thinkthat we have the same coefficient for all the three gases.

- L141: What about b_O3?

- L167: "dependence [in] particle size"(not [pn])

- L167: I cannot understand the whole sentence. Do you mean: "high dependence in particle size [distribution] THROW the aerosol phase function?

- Figure 4.: In the legend, maybe mention that P has no unit and also mention that P*L (figure on the right) is in W.m-2.sr-1 (like L). If not, the reader has to guess it from L-graphic and P-graphic.

- L231 (Equation 13): Are you sure? I would write: DNI_corr = DNI_measured – CSR = DNI_SUN_estimation

- L239 You define the CR (Circumsolar Ratio). Please write the equation that defines it as Equation 15

- L248 You cite Equation 15 that does not exist (surely it was your intention that Eq 15 is the definition of CR but you forget it)

- Figure 7 (Legend): "at at"

- Table 4: It is unclear regarding the table, which columns are with and witch columns are without CSR correction, since it is written "CSR Unc." everywhere. I guess that in each column pair, left is without and right is with correction, but please correct the header.

- L271 "good agreement", maybe you should here define what you consider being a "good agreement", by mentioning WMO traceability criteria that is cited below (L304).

- L290 340 nm. Instrumental uncertainty only? Maybe also because Rayleigh is higher and also aerosol scattering is higher -> Same comment for discussion in L311-L312

- L291: "model characterization [in] this range"

- L298: "MB >= -1.6 %" this is confusing, please discuss in absolute: "abs(MB) <= 1.6 %"

References:

For WMO Reports, please cite the page, at least the part of these very large reports in which the information is, in order to help the reader to find the relevant information for this study.

- L570 (Reference WMO, 1986): "GAW Report-No. 43" (not "437").
* * *

---

## Referee Comment (RC2) · Anonymous Referee #2 · 17 Jan 2020

The current work presents AOD retrievals form EKO MS-711, compared with CIMEL retrievals at Izaña Observatory and most importantly proposes an approach to correct DNI in respect to different FOV of the instruments using CSR. The paper fits perfectly the purposes of AMT and the proposed correction could find greater use in a number of instruments. Details of the approach are well presented and described sufficient in order to be repeatable. Results presented fortify the validity of the approach and are a guide for future studies of other spectroradiometers. The structure of the presentation is very steady and bibliographical review of the subject is more than sufficient. I suggest

the acceptance of the article for publication at AMT, after some minor corrections and clarifications.

More specifically

L22. This sentence seems a little poor and inadequate. I suggest to restate.

Paragraph 2.2 I think some information on the measuring schedule should be added. There is one spectra per minute or they are multiple spectra averaged and stored per minute? Exposure time is steady or it is changed according to the intensity of the irradiance? Are there oversaturation problems? Are there any filters used?

L105 Level 1.5 are automatic cloud screening and the quality assured data are L2.0. Please restate to be clear

L. 155 Have you used O3 in the calculations? There is nothing about it and at least for 340nm is important. If you have not calculated ozone absorption probably it could explain a part of the differences at 340 nm. L166-167 Please restate this sentence because it is not clear.

Paragraph 3.2 The measured spectrum has a resolution of 0.4nm with FWHM of 7 nm. When referring to monochromatic retrievals of AOD, have you used just one channel (which?_) or do you have convoluted multiple channels to a slit function? Please clarify this because it is crucial for understanding the differences with AERONET. For example lines 293-295 confused me on this matter. Also, I think it should be cleared if there any other difference with AERONET calculations (air masses, Rayleigh etc).

L248 There is no equation 15 in the manuscript

Paragraph 3.4 I understand that dust aerosols are the main in Izaña, but I think it is important to add some discussion of potential differences for other aerosol types.

Table 4. There is typo and all columns seem to be uncorrected.

L296-298. Please refer the number of datapoints used for each of the two periods.

L.3131 Also refer the number of data with AOD>0.1

---

## Referee Comment (RC3) · Anonymous Referee #3 · 18 Jan 2020

Review of the "Characterization of an EKO MS-711 spectroradiometer: aerosol retrieval from spectral direct irradiance measurements and corrections of the circumsolar radiation."

The paper presents results of direct sun measurements and aerosol optical depth (AOD) retrieval for an EKO MS-711 spectroradiometer. An extended investigation is presented for the circumsolar radiation correction.

In my opinion the paper is very well written and is well within the scope of AMT. Spectroradiometers have been used less nowadays for atmospheric monitoring due to reasons that the authors quote in their manuscript and I personally agree. However, they are very important instrumentation as the spectral characteristics of the solar irradiance is the desired one in order to be used for a number of atmospheric-radiation related issues.

I only have some minor comments on the manuscript.

Instrument characterization and performance.

The authors use the term instrument characterization in the title so I would expect some results on other aspects such as linearity, stray light etc.

In their instruments characteristics table they quote that the instrument step is way less (∼20 times) that the optical resolution. Can you provide some more information on how each measurement is performed? is it some kind of averaging ? or just a very wide entrance slit ?

The fact that the optical resolution is ∼7nm compared with 2nm and 4nm for CIMEL UV bands (I had the impression that CIMEL 380nm filters are also 2 nm wide), could be a source of uncertainties in the Rayleigh or Langley constants parameters of the EKO compared with the CIMEL? Meaning that the spectrum relative changes for differ-ent solar angles and atmospheric conditions can be different for irradiances at 340nm ±7nm and 340nm ±2nm.

The calibration constants and difference with the manufacturer ones seems noisy in the UV range, authors claim that "differences are attributed to the low halogen lamp signal in this region experienced during the factory calibration, and low instrument sensitivity in this region" could this affect AOD at UV results?

However, the stability of the instrument in the visible+ range for the 3 year period be-tween the manufacturer and the Langley calibrations are impressive. Maybe this also has to be pointed out in the text.

Circumsolar radiation

Circumsolar radiation contribution to the "true" measured direct irradiance is linked with AOD and also with aerosol types (phase functions). Higher AODs and forward scattering aerosols would introduce higher circumsolar correction factors. As in this work it is mentioned that a mixed (OPAC) based aerosol type is used, have you tested the actual correction and the effect on the AOD retrievals on a day with very high AOD and forward scattered aerosol type (e.g. dust aerosols) ?

Other Line 41 GAW-PFR showing lower values Table 1 : cosine response : is that applicable to the DNI spectral measurements ? Lines 108-110 : is this for direct or global irradiance ? Lines 212: 0.09"

Congratulations for a very interesting work.

---

## Author Comment (AC1) · 31 Mar 2020

*Referee #1: Lionel Doppler*

GENERAL COMMENTS

This paper presents a method to retrieve aerosol optical depth (AOD) out of spectral DNI (direct sun normal irradiance) radiation measurements from the spectroradiometer EKO MS-711. The paper presents the instrument, the site of the observations (IZO: Izaña Atmospheric Observatory), and the method used. An issue that is well discussed is how to correct the measured DNI, obtained with the EKO instruments that has a larger field of view than the WMO standards suggest for AOD measurements. The solution found is to estimate the CSR (circumsolar radiation) by simulating the forwarded scattered radiation with a radiative transfer code and multiplying it with a so-called penumbra function depending on the solar angles (azimuth and zenith). The method of AOD inversion is validated thanks to a comparison to a reference instrument (the Cimel – Aeronet photometer) for six wavelengths in UVA, VIS and NIR at the site of IZO during four months (April – July 2019). A statistical study is presented to validate the AOD retrieval method and evaluate the gains of the CSR correction.

The most innovative part of the paper is the presentation of the CSR estimation and the correction of the DNI for this instrument having a field of view of 5° in order to be compared to photometers having a field of view of less than 1.2°(WMO standards). This method is well explained in the paper and the reader can be convinced of the reliability of it.

The main concept presented in the paper is the AOD retrieval out of spectral DNI measurements from a spectroradiometer, this is not new, but only few articles are making a detailed presentation of the method explaining each step and showing all the equations. This is well done in this paper and will be useful for the AOD community, the photometer community and the spectroradiometer community.

The validation of the method is shown thanks to a detailed statistic comparison to a reference instruments, mentioning WMO traceability criteria and discussing objectively, fairly and humbly the weak points of the method and instrument. Thus, substantial conclusions are reached: the paper evaluates quantitatively the DNI correction method, the AOD retrieval method and its application to the instrument EKO MS-711, convincing the readers that these methods can be used operationally with this instrument.

The scientific methods used are well described their validity are discussed, a good balanced use of figures and mathematic equations contributes to a clear outline of them.

The references list is complete enough giving proper credit to current and past work related to this topic. The number of references is good balanced and the references are of excellent quality. Thanks to this literature work, the authors could clearly put forward their own contribution to the topics approached in this paper.

The title of the paper reflects the content of the paper in a good way; the abstract is a good complement of the title and a concise and truth summary of the paper.

The overall presentation is well structured, and despite some minor details (to which I suggested improvements in the part below named "technical comments") clear expressed.

The language is fluent and precise and it is an obstacle neither to get rapidly a good comprehensive view of this work nor to understand the technical and mathematical details. The mathematical formulae are shown in a good way. The equations are correct written, without mistake and well understandable.

I would suggest some minor improvements to be done: A table with all acronyms would be welcome. Also, I join a list of technical corrections (see below: "technical comments"). Moreover, some points should be briefly discussed, these questions are asked below in "specific comments". These are minor/technical corrections that I suggest.

Despite these technical corrections that have do be done, the article is of good scientific quality, of good significance and of good presentation. This justifies my evaluation here above and the fact that I suggest the editor to accept the manuscript and to ask for technical corrections and to answer to the four questions mentioned here below in "specific comments / questions"

*Authors: We appreciate the positive and constructive comments. Below we answer Dr. Doppler's comments.*

**2. SPECIFICT COMMENST / QUESTIONS**

C1.- About the CSR correction presented in 3.4.: The simulated forwarded scattering radiation is computed using desert dust aerosol. How can we adapt the correction factors to other type of aerosols? And if it is possible: How is it possible to integrated the characterization of the aerosol kind in an operational algorithm in order to have directly the CSR correction factors suitable to the defined aerosol type?

*Authors: The correction proposed in this work can be adapted to other types of aerosol- mixtures or sites. We have only used dust since at Izaña Observatory only two very contrasting situations are normally present: clean atmosphere with almost no aerosols, or dusty conditions under Saharan intrusions, mainly in summer. So, the correction factor has been specifically determined for dust aerosol.*

*Any way, we have included in the paper the following information from LibRadtran simulations that can be used for other types of aerosols. Apart from the graph, we have included in the Appendix B, a table with simulated CR values as a function of AOD for different types of aerosols. These values could be used in an operational AOD correction formula.*

*We have added this information in the final manuscript as follows:*

> *"…These results have been simulated considering the typical conditions of IZO where mineral dust is practically the only aerosol present (Berjón et al., 2019; García et al., 2017). Simulations of the effect on CR of the eight OPAC mixture aerosols available in LibRadtran model, continental (clean, average and polluted), urban, maritime (clean, polluted and tropical) and desert aerosols (Hess et al., 1998), and for a FOV=5°, are shown in Figure 6. For SZA=30°, with an $AOD_{500nm}$ range between 0 and 2 at sea level,*

*two defined groups are distinguished: the continental and urban aerosol mixtures, and the maritime and desert dust mixtures. It should be noted that for stations located in urban or continental (clean and contaminated) environments, which are the majority, the correction that would have to be made to the AOD for a very high aerosol load (e.g., AOD = 1) would be much lower, between 1/3 to 1/6, than the correction that would have been performed in the case of dust aerosol. (Figure 6 and Appendix B)."*

[Figure]

*Figure 6. Simulations of CR (%) for SZA 30° at sea level for AOD values between 0 and 2, at 500 nm, for different types of aerosols for FOV of 5°.*

*The following references have been added:*

*Berjón, A., Barreto, A., Hernández, Y., Yela, M., Toledano, C., and Cuevas, E.: A 10-year characterization of the Saharan Air Layer lidar ratio in the subtropical North Atlantic, Atmos. Chem. Phys., 19, 6331-6349, https://doi.org/10.5194/acp-19-6331-2019, 2019.*

*García, M. I., Rodríguez, S., and Alastuey, A.: Impact of North America on the aerosol composition in the North Atlantic free troposphere, Atmos. Chem. Phys., 17, 7387-7404, https://doi.org/10.5194/acp-17-7387-2017, 2017.*

*Hess, M., Koepke, P., and Schult, I.: Optical properties of aerosols and clouds: The software package OPAC, B. Am. Meteorol. Soc., 80, 831–844, 1998.*

*We have added the following table in the Appendix B with the numbers plotted in Figure 6.*

*(Table of Appendix B) **Numerical values of the CR (%) simulations for SZA 30° at sea level for AOD values between 0 and 2, at 500 nm, for different types of aerosols for FOV of 5°.***

| AOD | Continental Clean CR (%) | Continental Average CR (%) | Continental Polluted CR (%) | Urban CR (%) | Maritime Clean CR (%) | Maritime Polluted CR (%) | Maritime Tropical CR (%) | Desert CR (%) |
|---|---|---|---|---|---|---|---|---|
| 0.1 | 0.3 | 0.2 | 0.1 | 0.1 | 0.6 | 0.5 | 0.6 | 0.6 |
| 0.2 | 0.5 | 0.4 | 0.3 | 0.3 | 1.3 | 1.0 | 1.2 | 1.3 |
| 0.3 | 0.7 | 0.6 | 0.4 | 0.4 | 1.9 | 1.5 | 1.9 | 1.9 |
| 0.4 | 1.0 | 0.8 | 0.6 | 0.5 | 2.5 | 2.0 | 2.5 | 2.5 |
| 0.5 | 1.3 | 1.0 | 0.7 | 0.7 | 3.2 | 2.5 | 3.1 | 3.1 |
| 0.6 | 1.5 | 1.2 | 0.9 | 0.8 | 3.8 | 3.1 | 3.7 | 3.8 |
| 0.7 | 1.8 | 1.4 | 1.0 | 0.9 | 4.5 | 3.6 | 4.4 | 4.4 |
| 0.8 | 2.0 | 1.6 | 1.2 | 1.1 | 5.1 | 4.1 | 5.0 | 5.0 |
| 0.9 | 2.3 | 1.8 | 1.3 | 1.2 | 5.8 | 4.6 | 5.7 | 5.7 |
| 1 | 2.6 | 2.0 | 1.5 | 1.3 | 6.5 | 5.2 | 6.3 | 6.3 |
| 1.1 | 2.9 | 2.2 | 1.7 | 1.5 | 7.1 | 5.7 | 7.0 | 7.0 |
| 1.2 | 3.2 | 2.4 | 1.8 | 1.6 | 7.8 | 6.3 | 7.6 | 7.6 |
| 1.3 | 3.5 | 2.7 | 2.0 | 1.8 | 8.5 | 6.8 | 8.3 | 8.3 |
| 1.4 | 3.8 | 2.9 | 2.2 | 2.0 | 9.2 | 7.4 | 9.0 | 8.9 |
| 1.5 | 4.1 | 3.2 | 2.4 | 2.1 | 9.9 | 8.0 | 9.7 | 9.6 |
| 1.6 | 4.4 | 3.4 | 2.6 | 2.3 | 10.6 | 8.5 | 10.4 | 10.3 |
| 1.7 | 4.7 | 3.7 | 2.8 | 2.4 | 11.4 | 9.1 | 11.1 | 10.9 |
| 1.8 | 5.1 | 3.9 | 3.0 | 2.6 | 12.1 | 9.7 | 11.8 | 11.6 |
| 1.9 | 5.4 | 4.2 | 3.2 | 2.8 | 12.8 | 10.3 | 12.5 | 12.3 |
| 2 | 5.8 | 4.5 | 3.4 | 3.0 | 13.6 | 10.9 | 13.2 | 13.0 |

**C2.- IZO is a site of low aerosol amount. The results presented in the statistical study to validate the method (part 4.) shows AOD ranging between 0.0 And 0.2 (eg: Figure 9). How many points of comparison do you have for AOD > 0.15? What do you expect it should happen for other sites having larger AOD (continental sites in middle Europe or close-urban areas)?**

*Authors:    Between 83% and 85% of the data correspond to AOD ≤ 0.15 for all wavelengths, while for AOD > 0.15 we have between 13% and 15% of the data in the period April-September 2019 at IZO (see Figure 1).*

[Figure]

*Figure 1.- Frequency of occurrence of Cimel-AOD at all wavelengths between April and September 2019 at IZO.*

*Considering that our dusty condition threshold value is 0.1, we have added the following information in the final manuscript:*

> *"… The improvement in AOD for AOD>0.1 conditions **(20% of the data for 340 and 380 nm, and 16% for the rest of the wavelengths)** is remarkable, as already mentioned in the CSR correction section. The scatter is also significantly reduced for all wavelengths and aerosol loads…"*

*According to Figure 6, for continental polluted and urban aerosols the circumsolar radiation is much lower that for dust, so the required AOD corrections should be much lower. The AOD correction for continental pollution and urban aerosols decreases the higher the AOD, faster that makes the correction for dust.*

**C3. Are the results shown in Part 4 restricted to cases with desert dust aerosols? If yes, do you have some preliminary results for other kinds of aerosols? What do you expect it should happens? If no (= the results shown corresponds to different mixtures and kinds of aerosols), do you have some differences between different kinds of aerosols detected?**

*Authors: Please, see reply to the comment C1*

**C4. The AOD retrieval method presented in 3.1 and 3.2 is well described. Nevertheless, I would discuss two points more in detail: 1) Do you take the same airmass for aerosols, water vapour, mixed gases and ozone? 2) How do you compute Rayleigh optical depth? With which formula (Bodhaine?) and with which values of the air pressure (Aeronet uses a 6 hours average taken from a model)?**

*Authors:*

1) *We have not used the same airmass for all the components. We have used the following equations:*

   - *Aerosols and water vapour:* $m_a \approx m_{h2o} = \dfrac{1}{cos(\theta) + 0.0548\,(92.65 - \theta)^{-1.452}}$ *(Kasten, 1966); where θ Is the solar zenital angle.*

   - *NO$_2$:* $m_{NO2} = \dfrac{1}{sin\,(\theta) + 602.30\,(90 - \theta)^{0.5}(27.96 + \theta)^{-3.4536}}$ *(Gueymard, 1995)*

   - *Ozone:* $m_{o3} = \dfrac{R+h}{\sqrt{(R+h)^2 - (R+r)^2\,sin^2(\theta)}}$ *(Komhyr, 1989); where R (6370 km) is the mean radius of the earth, r is the station height above mean see level in km, and h is the mean height of the ozone layer in km (22 km).*

   - *Rayleigh and oxygen:* $: m_R \approx m_{o2} = \dfrac{1}{cos(\theta) + 0.50572\,(96.07995 - \theta)^{-1.6364}}$ *(Kasten and Young, 1989; Gueymard, 2001 )*

2) *The Rayleigh optical depth has been calculated from the following equation:*
   $\tau_R = \dfrac{P}{Po}\,0.008569\lambda^{-4}(1 + 0.0113\lambda^{-2} + 0.00023\lambda^{-4})$ *(Hansen and Travis, 1974); where P is the pressure at the measurement site within the earth's atmosphere in KPa, Po is the standard pressure at sea level and λ is the wavelength in μm.*
   *We have used in-situ pressure values used to determine $\tau_R$ at the same time the spectra were measured..*

*These equations have been added in the Sect. 3.1 and 3.2 as follow:*

"...

$$\tau(\lambda) = \tau_R(\lambda) + \tau_a(\lambda) + \tau_{NO_2}(\lambda) + \tau_{H_2O}(\lambda) + \tau_{O_2}(\lambda) + \tau_{O_3}(\lambda) \qquad (2)$$

where $\tau_R(\lambda)$ is the Rayleigh optical depth (**Hansen and Travis, 1974**) due to the molecular scattering that depends on the station pressure as well as on the optical air mass ($m_R$) (Kasten and Young, 1989), $\tau_a(\lambda)$ is the AOD, and the rest of the terms are the absorption by atmospheric gases in the affected wavelengths (Gueymard, 2001), which are defined as follows:

$$\tau_R = \frac{P}{Po} 0.008569\lambda^{-4}(1 + 0.0113\lambda^{-2} + 0.00023\lambda^{-4}) \qquad (3)$$

**where P is the pressure at the measurement site within the earth's atmosphere, Po is the standard pressure at sea level and $\lambda$ is the wavelength in µm. In-situ actual pressure at IZO was used.**

$$\tau_{NO_2} = u_{NO_2} A_{NO_2} \qquad (4)$$

where $u_{NO_2}$ is the reduced path-length (in atm-cm) taken from the OMI total column NO2 monthly average climatology and $A_{NO_2}$ its spectral absorption coefficient (Rothman et al., 2013).

$$\tau_{H_2O} = (U_{H_2O} A_{H_2O})^{b_{H2o}} \qquad (5)$$

where $U_{H_2O}$ is the column water vapour content (precipitable water) taken from a Global Navigation Satellite System (GNSS) receiver considering satellite precise orbits at IZO (Romero Campos et al., 2009), $A_{H_2O}$ the spectral absorption coefficient Rothman et al. (2013), and the $b_{H2o}$ exponent depends on the central wavelength position, instrument filter function, as well as the atmospheric pressure and temperature (Halthore et al., 1997). We have determined $\tau_{H_2O}$ from the transmittance for different water vapour and solar zenith angle (SZA) values from the MODTRAN model (Raptis et al., 2018).

$$\tau_{O_2} = (U_{O_2} A_{O_2})^{b_{O2}} \qquad (6)$$

where $U_{O_2}$ is the altitude-dependent gaseous scaled path-length taken from the Fourier transform infrared spectrometer (FTIR) measurements at IZO (Schneider et al., 2005), $A_{O_2}$ is the spectral absorption coefficient (Rothman et al., 2013), and the $b_{O2}$ exponent was obtained from the transmittance values simulated with the MODTRAN model (Berk et al., 2000) for IZO, obtaining a value of 0.454. This value is similar to that obtained by Pierluissi and Tsai (1986, 1987).

$$\tau_{O_3} = U_{O_3} A_{O_3} \qquad (7)$$

where $U_{O_3}$ is the total column ozone obtained with a reference Brewer spectrophotometer at IZO (Redondas et al., 2018), and $A_{O_3}$ is the ozone absorption cross section (Brion et al., 1993, 1998).

*The Langley-Plot determines $DNI_o(\lambda)$ (that allows to derive calibration constant) from a linear extrapolation of $DNI(\lambda)$ measurements to zero air mass, corrected to mean Sun–Earth distance, and plotted on a logarithmic scale versus air mass:*

$$ln\big(DNI(\lambda)\big) = ln\ DNI_o\ (\lambda) - [\tau_R\ (\lambda)m_R + \tau_a(\lambda)m_a + \tau_{NO_2}(\lambda)m_{NO_2} + \tau_{H_2O}(\lambda)m_{H_2O} + \tau_{O_2}(\lambda)m_{O_2} + \tau_{O_3}(\lambda)m_{O_3}] \qquad (8)$$

*where the different air masses have the following expressions:*

$$m_R \approx m_{o2} = \frac{1}{cos(\theta)+0.50572\ (96.07995-\theta)^{-1.6364}}; \quad \text{(Kasten and Young, 1989; Gueymard, 2001)} \qquad (9)$$

$$m_a \approx m_{h2o} = \frac{1}{cos(\theta)+0.0548\ (92.65-\theta)^{-1.452}}; \text{(Kasten, 1966 )} \qquad (10)$$

$$m_{NO2} = \frac{1}{sin\ (\theta)+602.30\ (90-\theta)^{0.5}(27.96+\theta)^{-3.4536}}; \text{(Gueymard, 1995)} \qquad (11)$$

$$m_{o3} = \frac{R+h}{\sqrt{(R+h)^2-(R+r)^2\ sin^2(\theta)}}; \text{(Komhyr, 1989)} \qquad (12)$$

**where R (6370 km) is the mean radius of the Earth, r is the station height above mean see level in km, and h is the mean height of the ozone layer in km (22 km).**

*…"*

**Section 3.2**

*"…*

*The AOD retrievals have been calculated from Eq. 8, as follows:*

$$AOD = \frac{1}{m_a}\Big[ln\ DNI_o(\lambda) - ln\ DNI(\lambda) - \tau_R\ (\lambda)m_R - \tau_a(\lambda)m_a - \tau_{NO_2}(\lambda)m_{NO_2} - \tau_{H_2O}(\lambda)m_{H_2O} - \tau_{O_3}(\lambda)m_{O_3}\Big] \qquad (14)$$

*Grouping the gases contributions as $\tau_{gas}$, the AOD expression is reduced to:*

$$AOD = \frac{1}{m_a}\Big[ln\ DNI_o(\lambda) - ln\ DNI(\lambda) - \tau_R\ m_R - \tau_{gas}\ m_{gas}\Big] \qquad (10)$$

*…"*

**3. TECHNICAL COMMENTS**

**General comment: Please introduce a list of all acronyms used**

*Authors: We have included all acronyms used in the manuscript in Appendix A.*

**Abstract:**

**- At the beginning of the abstract, should be explained what is the spectral range and resolution of EKO MS-711.**

*Authors: We have included the spectral range of the spectroradiometer in the abstract as follows:*

*"Spectral direct UV-Visible normal solar irradiance (DNI) has been measured with an EKO MS-711 **grating** spectroradiometer, **which has a spectral range of 300-1100 nm, 0.4 nm step,** at the Izaña Atmospheric Observatory (IZO, Spain) … "*

**Introduction**

**- L25: "properties, such as single scattering albedo, size distribution, etc" -> please avoid "etc", write a complete list, best sorted in decreasing importance order.**

*Authors:  We have completed this sentence as follows:*

> *"… therefore it is necessary to make more efforts to evaluate the aerosol atmospheric content and optical properties, such as the **aerosol optical depth (AOD), Angström exponent (AE), single scattering albedo (SSA), scattering coefficient, and absorption coefficient.**"*

**- L47: (again etc.) -> please complete list or use "e.g.:"**

*Authors: We have modified this sentence as follows:*

> *"… possibility to provide other atmospheric components **(e.g., $O_3$, $NO_2$, $SO_2$, $CH_4$, and $H_2O$)**…"*

**- L47: reference is Barreto 2014 (and not 2013) for spectroradiometer and Aerosol.**

*Authors:  You are right. The reference will be replaced by Barreto et al. (2014) in the final manuscript.*

**- L67: Go to next line before presenting the parts of your papers with "We have devided:"**

*Authors:  Done*

**Part 2: Site Description, Instrument and ancillary information**

**2.2 Instrument: Maybe explain what kind of technology it is: monochromator or array spectrometer (it is not specified).**

*Authors: We have added the instrument type as follows:*

> *"… An EKO MS-711 **grating** spectroradiometer used in direct-sun measurement mode has been tested…"*

**- L98: Specify in this part of the text that the world AOD reference is the PFR in order that the reader knows from which instrument you are talking about.**

*Authors:  We have addressed this issue  as follows:*

> *"…The different Cimel references have been shown to have a good AOD traceability with the **GAW-PFR worldwide reference** (Cuevas et al., 2019)…"*

**- L104: Bias < 0.01 (and not > 0.01) (citing Sinyuk, GRL 2012)**

*Authors: Done, it was a typo.*

**Part 3. Methodology**

**- L130+L136+L141, maybe use a different description for "b" of each gas: b_H2O, b_O2, b_O3 for example. Here you have the same letter and the reader can think that we have the same coefficient for all the three gases.**

*Authors: Done in the final manuscript (see question 4:  SPECIFICT COMMENST / QUESTIONS)*

**- L141: What about b_O3?**

*Authors:  It has been corrected in the manuscript final (see reply to comment C4)*

**- L167: "dependence [in] particle size"(not [pn])**

*Authors:  Done. It was a typo.*

**- L167: I cannot understand the whole sentence. Do you mean: "high dependence in particle size [distribution] THROW the aerosol phase function?**

*Authors: We have clarified this sentence as follows:*

> *"… This CSR has a high dependence **on the particle size (Räisänen and Lindfors, 2019), thus large particles (such as desert dust) produce a higher  scattering on the incident beam than the smaller particles (e.g., rural background aerosols), leading this contribution to overestimate the DNI…"***

**- Figure 4.: In the legend, maybe mention that P has no unit and also mention that P*L (figure on the right) is in W.m-2.sr-1 (like L). If not, the reader has to guess it from L-graphic and P-graphic.**

*Authors: We have added the units in the legend as follows:*

[Figure]

*Figure 4. Example of the  (a) diffuse radiance L (Wm-2µm-2sr-2) at 500 nm, shown in colours for different SZA and φ ; (b) penumbra function P determined from Eq. 11  and (c) the product of the diffuse radiance L and penumbra function P.*

**- L231 (Equation 13): Are you sure? I would write: DNI_corr = DNI_measured – CSR= DNI_SUN_estimation**

*Authors:  You are right, the equation has been modified in the final manuscript as follows:*

$$DNI_{CORR} = DNI - CSR$$

**- L239 You define the CR (Circumsolar Ratio). Please write the equation that defines it as Equation 15**

*Authors:  The equation that defines CR has been added in the final manuscript as follows:*

$$CR(\%) = \frac{CSR}{DNI_{SUN} + CSR} \cdot 100$$

**- L248 You cite Equation 15 that does not exist (surely it was your intention that Eq 15 is the definition of CR but you forget it)**

*Authors:  You are right. We forgot it.*

**- Figure 7 (Legend): "at at"**

*Authors:  Done*

**- Table 4: It is unclear regarding the table, which columns are with and witch columns are without CSR correction, since it is written "CSR Unc." everywhere. I guess that in each column pair, left is without and right is with correction, but please correct the header.**

*Authors:  Done, the table has been corrected as follows:*

| Wavelength (nm) | R | | Slope | | RMS | | MB | |
|---|---|---|---|---|---|---|---|---|
| | CSR Unc. | CSR Corr. | CSR Unc. | CSR Corr. | CSR Unc. | CSR Corr. | CSR Unc. | CSR Corr. |
| 340 nm | 0.960 | 0.973 | 1.063 | 0.994 | 0.017 (28.9%) | 0.007 (16.9%) | 0.015 (24.5%) | <0.001 (-1.4%) |
| 380 nm | 0.981 | 0.986 | 1.071 | 1.001 | 0.009 (20.2%) | 0.005 (12.9%) | 0.007 (14.8%) | <0.001 (1.2%) |
| UV-Range | 0.971 | 0.979 | 1.067 | 0.997 | 0.013 (24.6%) | 0.006 (14.9%) | 0.011 (19.7%) | <0.001 (1.3%) |
| 440 nm | 0.984 | 0.987 | 1.041 | 0.997 | 0.101 (22.4%) | 0.005 (13.5%) | 0.009 (18.7%) | 0.001 (0.6%) |
| 500 nm | 0.988 | 0.991 | 1.075 | 1.018 | 0.007 (18.2%) | 0.005 (12.9%) | 0.004 (12.1%) | 0.002 (0.4%) |
| 675 nm | 0.989 | 0.991 | 1.057 | 1.013 | 0.006 (19.7%) | 0.006 (10.7%) | 0.003 (11.2%) | <0.001 (0.5%) |
| 870 nm | 0.998 | 0.999 | 1.039 | 1.009 | 0.004 (18.8%) | 0.003 (7.3%) | <0.001 (0.3%) | <0.001 (0.2%) |
| VIS-Range | 0.989 | 0.992 | 1.053 | 1.009 | 0.029 (19.5%) | 0.005 (11.1%) | 0.004 (10.6%) | <0.001 (0.4%) |

*Table 4. Statistics of the comparison between EKO AOD, with no CSR corrections **(CSR Unc.)** and implementing CSR corrections **(CSR Corr.)**, and Cimel AOD at 340, 380, 440, 500, 675 and 870 nm at IZO between April and September 2019. R: correlation coefficient, slope of the least-squares fit between EKO AOD and Cimel AOD, RMS: root mean square of the bias and MB: mean bias. The results of the relative bias are in brackets (in %).*

**- L271 "good agreement", maybe you should here define what you consider being a "good agreement", by mentioning WMO traceability criteria that is cited below (L304).**

*Authors: In this case we are just comparing the AOD provided by the EKO and Cimel. Later, we compare the AOD of both instruments using the WMO traceability criteria. We have clarified what means "good agreement" by adding the correlation coefficient. This sentence reads now as follows:*

> *"… The results show (Table 4) that there is a good agreement **(correlation coefficient > 0.98)** between EKO AOD and Cimel AOD for all channels, even for no CSR correction…"*

**- L290 340 nm. Instrumental uncertainty only? Maybe also because Rayleigh is higher and also aerosol scattering is higher -> Same comment for discussion in L311-L312**

*Authors: The authors have attributed most of the found differences to the instrument uncertainty because the instrument error in the spectral range between 300 and 350 nm is 17.2%, of which 6% corresponds to stray-light, and 6% corresponds to measurement repeatability. Moreover, it is also affected by the different FWHM between EKO (7 nm) and CIMEL (2 nm) at 340 nm, and by the fact that Rayleigh and aerosol scattering are higher in the UV range (Cuevas et al., 2018).*

*We have added this information in the final manuscript as follows:*

*"… The scatter also is significantly reduced for all wavelengths and aerosol loads, except in the 340 nm UV channel. This is **mainly** attributed to **the instrumental error in the spectral range between 300 and 350 nm (17.2%), of which 6% corresponds to stray-light and 6% corresponds to measurement repeatability (Zong et al., 2006)**, to the **different FWHM between EKO (7 nm) and CIMEL (2 nm) at 340 nm, and to the fact that Rayleigh and aerosol scattering are higher in the UV range (Cuevas et al., 2019)…"***

**L291: "model characterization [in] this range"**

*Authors: Done*

**- L298: "MB >= -1.6 %" this is confusing, please discuss in absolute: "abs (MB) <= 1.6%"**

*Authors: Done*

**References:**

**For WMO Reports, please cite the page, at least the part of these very large reports in which the information is, in order to help the reader to find the relevant information for this study.**

*Authors: We have added this information in the final manuscript.*

**- L570 (Reference WMO, 1986): "GAW Report-No. 43" (not "437").**

*Authors: Done*

---

## Author Comment (AC2) · 31 Mar 2020

**Referee #2**

The current work presents AOD retrievals form EKO MS-711, compared with CIMEL retrievals at Izaña Observatory and most importantly proposes an approach to correct DNI in respect to different FOV of the instruments using CSR. The paper fits perfectly the purposes of AMT and the proposed correction could find greater use in a number of instruments. Details of the approach are well presented and described sufficient in order to be repeatable. Results presented fortify the validity of the approach and are a guide for future studies of other spectroradiometers. The structure of the presentation is very steady and bibliographical review of the subject is more than sufficient. I suggest the acceptance of the article for publication at AMT, after some minor corrections and clarifications.

Authors: We acknowledge the referee's positive and constructive comments. Below we respond to his/her general comments.

**General Comments:**

**More specifically**

L22. This sentence seems a little poor and inadequate. I suggest to restate.

Authors: We have modified the sentence as follows:

"One of the most important elements that governs the Earth's climate, and its processes, is the presence of atmospheric aerosols, which produce a significant radiative forcing resulting from light scattering and absorption, and radiation emission. Moreover, they act as cloud condensation nuclei, modifying cloud properties (IPCC, 2013). Aerosols effect on the Earth Radiation Balance has been quantified as a cooling of -0.45 W m-2, and -0.9 W m-2 when considering the combined effect of both aerosols and clouds..."

Paragraph 2.2 1) I think some information on the measuring schedule should be added. 2) There is one spectral per minute or they are multiple spectra averaged and stored per minute? 3) Exposure time is steady or it is changed according to the intensity of the irradiance? 4) Are there oversaturation problems? 5) Are there any filters used?

**Authors:**

1) and 2) The EKO MS-711 spectroradiometer measures one spectrum per minute.

*3)* The exposure time is not constant. The setting changes automatically according to the intensity of the irradiance, and varies from 10 ms to 5 s.

4) As a result of the optimized exposure time for the irradiance and the instrument measurement dynamic range, no saturated measurements are experienced.

5) This instrument does not use filters.

Followings the recommendations of the referee, we have added this information as follows:

**Section 2.2**

"An EKO MS-711 **grating** spectroradiometer used in direct sun measurement mode has been tested (Figure 1) within the CIMO Testbed program from April to September 2019 (14706 datapoints) ..."

"...This spectroradiometer has been mounted on an EKO sun-tracker STR-21G-S2 (accuracy of <0.01°). This setup performs one spectrum per minute, with an exposure time that changes automatically according to the intensity of the irradiance that varies from 10 ms to 5 s. The main specifications of the EKO MS-711 spectroradiometer are shown in Table 1..."

L105 Level 1.5 are automatic cloud screening and the quality assured data are L2.0. Please restate to be clear.

Authors: The authors have not used Level 2.0 because it is not available for the study period (April and September 2019) in AERONET. We have modified the sentence as follows:

"...In this study, we have used AERONET Version 3.0 Level 1.5 AOD data..."

L155 Have you used O3 in the calculations? There is nothing about it and at least for 340nm is important. If you have not calculated ozone absorption probably it could explain a part of the differences at 340 nm.

Authors: Yes. We have taken into account ozone column in the AOD retrievals at 340, 500 and 675 nm (see Table 2 of the manuscript). The ozone values used have been measured with a reference double Brewer spectroradiometer at Izaña station, therefore it does not explain the differences found at 340 nm.

*This information is given in the Section 3.1, however, the authors have added this information in the Section 3.2 as follows:*

"...In this work, we have calculated the EKO AOD at the same nominal wavelengths as those of the Cimel (340, 380, 440, 500, 675 and 870 nm) following the methodology used by AERONET (Holben et al. (2001); Giles et al. (2019), and references herein). For each wavelength, we have taken into account the spectral corrections shown in Table 2. All wavelengths have been corrected by the Rayleigh scattering (see Sect. 3.1). Furthermore the 340, 380, 440 and 500 nm are corrected from nitrogen dioxide (NO2) absorption, being the optical depth calculated using the OMI total column NO2 climatological monthly averages, and the NO2 absorption coefficient from Burrows et al. (1999).The 340, 500 and 675 nm channels are corrected of ozone, using the ozone values from the Izaña WMO-GAW reference Brewer spectroradiometer..."

L166-167 Please restate this sentence because it is not clear.

Authors: We have modified the sentence as follows:

"... This CSR has a high dependence on the particle size (Räisänen and Lindfors, 2019), thus large particles (such as desert dust) produce a higher scattering on the incident beam than the smaller particles (e.g., rural background aerosols), leading this contribution to overestimate the DNI..."

Paragraph 3.2 The measured spectrum has a resolution of 0.4nm with FWHM of 7 nm. When referring to monochromatic retrievals of AOD, have you used just one channel(which?) or do you have convoluted multiple channels to a slit function? Please clarify this because it is crucial for understanding the differences with AERONET. For example lines 293-295 confused me on this matter. Also, I think it should be cleared if there any other difference with AERONET calculations (air masses, Rayleigh etc).

Authors: For determining the AOD with the EKO MS-711 spectroradiometer, we have considered the same nominal wavelengths and bandwidths (Filter Bandpass) as those of the Cimel (340: 2 nm, 380: 4 nm, 440: 5 nm, 500: 5 nm, 675: 5 nm and 870: 5 nm) as indicated on Table 2 of the manuscript. Centred on each wavelength and with its corresponding bandwidth, we have performed the integration of the irradiance on the considered spectral range. For example, in the AOD retrieval at 500 nm, the range 495-505 nm is used to perform the integration:

$$DNI(\lambda) = \int_{495 nm}^{505 nm} DNI(\lambda)_{EKO-MS711} d\lambda$$

This integrated value is the one used in equations of paragraph 3.2.

We have modified this paragraph as follows:

"...In this work, we have calculated the EKO AOD at the same nominal wavelengths as those of the Cimel (340, 380, 440, 500, 675 and 870 nm), **by integrating the measured irradiance on the considered bandpass (see Table 2),** following the methodology used by AERONET (Holben et al. (2001); Giles et al. (2019), and references herein). **For each wavelength**, we have taken into account the spectral corrections shown in Table 2. All **wavelengths have been corrected by the Rayleigh scattering (see Sect. 3.1). Furthermore** the 340, 380, 440 and 500 nm channels have been corrected from nitrogen dioxide (NO2) absorption, being its optical depth calculated using the OMI total column NO2 climatological monthly averages, and the NO2 absorption coefficient from Burrows et al. (1999). **The 340, 500 and 675 nm channels have been corrected of ozone, using the ozone values from a GAW reference Brewer spectroradiometer sited at Izaña Observatory...**"

Regarding the Lines 293-295, maybe the confusion arises in the sentence "some additional radiation contribution from the adjacent wavelengths". The considered range on each channel are those explained before and, in the paragraph, we tried to highlight that for the UV channels the contribution of the stray-light is important, therefore we have modified the paragraph as follows:

"...Since the 340 nm and 380 nm channels have 2 nm and 4 nm bandpass, respectively, and the EKO MS-711 FWHM is 7nm (Table 1), these two UV channels have some additional radiation contribution from the adjacent wavelengths **due to stray-light**, increasing their uncertainty and causing an AOD overestimation..."

The equations of air masses and optical depths used are the same to those used by AERONET, and they have been included in the final manuscript.

**L248 There is no equation 15 in the manuscript**

Authors: Thank you. The equation of the CR has been added in the final manuscript as follows:

$$CR(\%) = \frac{CSR}{DNI_{SUN} + CSR} \cdot 100$$

Paragraph 3.4 I understand that dust aerosols are the main in Izaña, but I think it is important to add some discussion of potential differences for other aerosol types.

Authors: We have only used dust since at Izaña Observatory only two very contrasting situations are normally present: clean atmosphere with almost no aerosols, or dusty conditions under Saharan intrusions, mainly in summer. So, the correction factor has been specifically determined for dust aerosol.

Any way, we have included in the paper the following information from LibRadtran simulations that can be used for other types of aerosols. Apart from the graph, we have included in the Appendix B, a table with simulated CR values as a function of AOD for different types of aerosols.

We have added this information in the final manuscript as follows:

"... These results have been simulated considering the typical conditions of IZO where mineral dust is practically the only aerosol present (Berjón et al., 2019; García et al., 2017). Simulations of the effect on CR of the eight OPAC mixture aerosols available in LibRadtran model, continental (clean, average and polluted), urban, maritime (clean, polluted and tropical) and desert aerosols (Hess et al., 1998), and for a FOV=5°, are shown in Figure 6. For SZA=30°, with an  $AOD_{500nm}$  range between 0 and 2 at sea level, two defined groups are distinguished: the continental and urban aerosol mixtures, and the maritime and desert dust mixtures. It should be noted that for stations located in urban or continental (clean and contaminated) environments, which are the majority, the correction that would have to be made to the AOD for a very high aerosol load (e.g., AOD = 1) would be much lower, between 1/3 to 1/6, than the correction that would have been performed in the case of dust aerosol. (Figure 6 and Appendix B) ..."

Figure 6. Simulations of CR (%) for SZA 30° at sea level for AOD values between 0 and 2, at 500 nm, for different types of aerosols for FOV of 5°.

The following references have been added:

Berjón, A., Barreto, A., Hernández, Y., Yela, M., Toledano, C., and Cuevas, E.: A 10-year characterization of the Saharan Air Layer lidar ratio in the subtropical North Atlantic, Atmos. Chem. Phys., 19, 6331-6349, https://doi.org/10.5194/acp-19-6331-2019, 2019.

García, M. I., Rodríguez, S., and Alastuey, A.: Impact of North America on the aerosol composition in the North Atlantic free troposphere, Atmos. Chem. Phys., 17, 7387-7404, https://doi.org/10.5194/acp-17-7387-2017, 2017.

Hess, M., Koepke, P., and Schult, I.: Optical properties of aerosols and clouds: The software package OPAC, B. Am. Meteorol. Soc., 80, 831–844, 1998.

We have added the following table in the Appendix B with the numbers plotted in Figure 6.

(Table of Appendix B) Numerical values of the CR (%) simulations for SZA 30° at sea level for AOD values between 0 and 2, at 500 nm, for different types of aerosols for FOV of 5°.

|     | Continental | Continental | Continental | Urban  | Maritime | Maritime | Maritime | Desert     |
|-----|-------------|-------------|-------------|--------|----------|----------|----------|------------|
| AOD | Clean       | Average     | Polluted    |        | Clean    | Polluted | Tropical |            |
|     | CR (%)      | CR (%)      | CR (%)      | CR (%) | CR (%)   | CR (%)   | CR (%)   | CR (%)     |
| 0.1 | 0.3         | 0.2         | 0.1         | 0.1    | 0.6      | 0.5      | 0.6      | 0.6        |
| 0.2 | 0.5         | 0.4         | 0.3         | 0.3    | 1.3      | 1.0      | 1.2      | 1.3        |
| 0.3 | 0.7         | 0.6         | 0.4         | 0.4    | 1.9      | 1.5      | 1.9      | 1.9        |
| 0.4 | 1.0         | 0.8         | 0.6         | 0.5    | 2.5      | 2.0      | 2.5      | 2.5        |
| 0.5 | 1.3         | 1.0         | 0.7         | 0.7    | 3.2      | 2.5      | 3.1      | 3.1 |
| 0.6 | 1.5         | 1.2         | 0.9         | 0.8    | 3.8      | 3.1      | 3.7      | 3.8        |
| 0.7 | 1.8         | 1.4         | 1.0         | 0.9    | 4.5      | 3.6      | 4.4      | 4.4        |
| 0.8 | 2.0         | 1.6         | 1.2         | 1.1    | 5.1      | 4.1      | 5.0      | 5.0        |
| 0.9 | 2.3         | 1.8         | 1.3         | 1.2    | 5.8      | 4.6      | 5.7      | 5.7        |

| 1          | 2.6 | 2.0         | 1.5 | 1.3 | 6.5         | 5.2  | 6.3         | 6.3  |
|------------|-----|-------------|-----|-----|-------------|------|-------------|-------------|
| 1.1        | 2.9 | 2.2         | 1.7 | 1.5 | 7.1         | 5.7  | 7.0         | 7.0         |
| 1.2        | 3.2 | 2.4         | 1.8 | 1.6 | 7.8         | 6.3  | 7.6         | 7.6         |
| 1.3        | 3.5 | 2.7         | 2.0 | 1.8 | 8.5         | 6.8  | 8.3         | 8.3  |
| 1.4        | 3.8 | 2.9         | 2.2 | 2.0 | 9.2         | 7.4  | 9.0         | 8.9  |
| 1.5        | 4.1 | 3.2         | 2.4 | 2.1 | 9.9  | 8.0  | 9.7         | 9.6  |
| 1.6        | 4.4 | 3.4         | 2.6 | 2.3 | 10.6        | 8.5  | 10.4        | 10.3 |
| 1.7        | 4.7 | 3.7         | 2.8 | 2.4 | 11.4        | 9.1  | 11.1        | 10.9 |
| 1.8        | 5.1 | 3. 9 | 3.0 | 2.6 | 12.1 | 9.7  | 11.8        | 11.6        |
| 1.9 | 5.4 | 4.2         | 3.2 | 2.8 | 12.8        | 10.3 | 12.5        | 12.3        |
| 2          | 5.8 | 4.5         | 3.4 | 3.0 | 13.6 | 10.9 | 13.2 | 13.0        |

**Table 4. There is typo and all columns seem to be uncorrected.**

Authors: Done. The final table is the following:

| Wavelength | R     |       | Slope |       | RMS     |         | MB      |         |
|------------|-------|-------|-------|-------|---------|---------|---------|---------|
| (nm)       | CSR   | CSR   | CSR   | CSR   | CSR     | CSR     | CSR     | CSR     |
|            | Unc.  | Corr. | Unc.  | Corr. | Unc.    | Corr.   | Unc.    | Corr.   |
| 340 nm     | 0.960 | 0.973 | 1.063 | 0.994 | 0.017   | 0.007   | 0.015   | <0.001  |
|            |       |       |       |       | (28.9%) | (16.9%) | (24.5%) | (-1.4%) |
| 380 nm     | 0.981 | 0.986 | 1.071 | 1.001 | 0.009   | 0.005   | 0.007   | <0.001  |
|            |       |       |       |       | (20.2%) | (12.9%) | (14.8%) | (1.2%)  |
| UV-Range   | 0.971 | 0.979 | 1.067 | 0.997 | 0.013   | 0.006   | 0.011   | <0.001  |
|            |       |       |       |       | (24.6%) | (14.9%) | (19.7%) | (1.3%)  |
| 440 nm     | 0.984 | 0.987 | 1.041 | 0.997 | 0.101   | 0.005   | 0.009   | 0.001   |
|            |       |       |       |       | (22.4%) | (13.5%) | (18.7%) | (0.6%)  |
| 500 nm     | 0.988 | 0.991 | 1.075 | 1.018 | 0.007   | 0.005   | 0.004   | 0.002   |
|            |       |       |       |       | (18.2%) | (12.9%) | (12.1%) | (0.4%)  |
| 675 nm     | 0.989 | 0.991 | 1.057 | 1.013 | 0.006   | 0.006   | 0.003   | <0.001  |
|            |       |       |       |       | (19.7%) | (10.7%) | (11.2%) | (0.5%)  |
| 870 nm     | 0.998 | 0.999 | 1.039 | 1.009 | 0.004   | 0.003   | <0.001  | <0.001  |
|            |       |       |       |       | (18.8%) | (7.3%)  | (0.3%)  | (0.2%)  |
| VIS-Range  | 0.989 | 0.992 | 1.053 | 1.009 | 0.029   | 0.005   | 0.004   | <0.001  |
|            |       |       |       |       | (19.5%) | (11.1%) | (10.6%) | (0.4%)  |

**Table 4**. Statistics of the comparison between EKO AOD, with no CSR corrections **(CSR Unc.)** and implementing CSR corrections **(CSR Corr.)**, and Cimel AOD at 340, 380, 440, 500, 675 and 870 nm at IZO between April and September 2019. R: correlation coefficient, slope of the least-squares fit between EKO AOD and Cimel AOD, RMS: root mean square of the bias and MB: mean bias. The results of the relative bias are in brackets (in %).

**L296-298. Please refer the number of datapoints used for each of the two periods.**

Authors: We have added the number of datapoints used for each of the two periods as follows:

"... The linear AOD-correction equations were determined by using data measured from April 1st to July 31th 2019 **(69% of the data)** at Izaña Observatory (Table 5). The validation of these linear AOD-correction equations was performed using an independent period of data (between August 1st and September 30th 2019) **(31% of the data)**..."

**L.3131 Also refer the number of data with AOD>0.1**

Authors: We have added the number of AOD>0.1 as follows:

"When focusing the analysis on relatively high AOD (AOD> 0.10), we found that the percentage of AOD differences out of the WMO  $U_{95}$  limits were  $\approx 3.5\%$  (**0.8% of de data**) at 380 nm and 0.6% (**0.3% of the data**) at 870 nm..."

---

## Author Comment (AC3) · 31 Mar 2020

**Referee #3**

The paper presents results of direct sun measurements and aerosol optical depth (AOD) retrieval for an EKO MS-711 spectroradiometer. An extended investigation is presented for the circumsolar radiation correction.

In my opinion the paper is very well written and is well within the scope of AMT. Spectroradiometers have been used less nowadays for atmospheric monitoring due to reasons that the authors quote in their manuscript and I personally agree. However, they are very important instrumentation as the spectral characteristics of the solar irradiance is the desired one in order to be used for a number of atmospheric-radiation related issues.

I only have some minor comments on the manuscript.

*Authors: We appreciate the positive and constructive comments of the Referee. Below we respond to his/her general comments.*

**Instrument characterization and performance.**

The authors use the term instrument characterization in the title so I would expect some results on other aspects such as linearity, stray light etc.

*Authors: We fully agree. We have modified the title of the manuscript as follows:*

> *Title: "Aerosol retrievals from the EKO MS-711 spectral direct irradiance measurements and corrections of the circumsolar radiation"*

In their instruments characteristics table they quote that the instrument step is way less (≈20 times) that the optical resolution. Can you provide some more information on how each measurement is performed? is it some kind of averaging? or just a very wide entrance slit?

*Authors: The EKO MS-711 spectroradiometer measures one spectrum per minute. The exposure time is not constant, but the setting changes automatically the exposure time between 10 ms to 5 s, according to the intensity of the irradiance.*

*So, we have added this information as follows:*

*Section 2.2*

> *"…This spectroradiometer has been mounted on an EKO sun-tracker STR-21G-S2 (accuracy of <0.01°). **This setup performs one spectrum per minute, with an exposure time that changes automatically according to the intensity of the irradiance that varies***

*from 10 ms to 5 s. The main specifications of the EKO MS-711 spectroradiometer are shown in Table 1…"*

**The fact that the optical resolution is ≈7nm compared with 2nm and 4nm for CIMEL UV bands (I had the impression that CIMEL 380nm filters are also 2 nm wide), could be a source of uncertainties in the Rayleigh or Langley constants parameters of the EKO compared with the CIMEL? Meaning that the spectrum relative changes for different solar angles and atmospheric conditions can be different for irradiances at 340nm ±7nm and 340nm ±2nm.**

*Authors:  For determining the AOD with the EKO MS-711 spectroradiometer, we have considered the same nominal wavelengths and bandwidths (Filter Bandpass) as those of the Cimel (340: 2 nm, 380: 4 nm, 440: 5 nm, 500: 5 nm, 675: 5 nm and 870: 5 nm) as indicated on Table 2 of the manuscript. Centred on each wavelength and with its corresponding bandwidth, we have performed the integration of the irradiance on the considered spectral range. For example, in the AOD retrieval at 500 nm, the range 495-505 nm is used to perform the integration:*

$$DNI(\lambda) = \int_{495\ nm}^{505\ nm} DNI(\lambda)_{EKO-MS711} d\lambda$$

*This integrated value is the one used in equations of paragraph 3.2.*

*We have modified this paragraph as follows:*

> *"…In this work, we have calculated the EKO AOD at the same nominal wavelengths as those of the Cimel (340, 380, 440, 500, 675 and 870 nm), **by integrating the measured irradiance on the considered bandpass (see Table 2),** following the methodology used by AERONET (Holben et al. (2001); Giles et al. (2019), and references herein). **For each wavelength**, we have taken into account the spectral corrections shown in Table 2. **All wavelengths have been corrected by the Rayleigh scattering (see Sect. 3.1). Furthermore** the 340, 380, 440 and 500 nm channels have been corrected from nitrogen dioxide ($NO_2$) absorption, being its optical depth calculated using the OMI total column $NO_2$ climatological monthly averages, and the $NO_2$ absorption coefficient from Burrows et al. (1999). **The 340, 500 and 675 nm channels have been corrected of ozone, using the ozone values from a GAW reference Brewer spectroradiometer sited at Izaña Observatory..."***

**The calibration constants and difference with the manufacturer ones seems noisy in the UV range, authors claim that "differences are attributed to the low halogen lamp signal in this region experienced during the factory calibration, and low instrument sensitivity in this region" could this affect AOD at UV results?**

*Authors:  Yes, it affects to AOD uncertainty in the UV range. We have clarified this issue as follows:*

> *"… The scatter also is significantly reduced for all wavelengths and aerosol loads, except in the 340 nm UV channel. This is **mainly** attributed to **the instrumental error in the spectral range between 300 and 350 nm (17.2%), of which 6% corresponds to stray-light and 6% corresponds to measurement repeatability (Zong et al., 2006), to the different FWHM between EKO (7 nm) and CIMEL (2 nm) at 340 nm, and to the fact that Rayleigh and aerosol scattering are higher in the UV range (Cuevas et al., 2019)…"***

However, the stability of the instrument in the visible+ range for the 3 years period between the manufacturer and the Langley calibrations are impressive. Maybe this also has to be pointed out in the text.

*Authors: We have added this information in the final manuscript as follows:*

> *"…The comparison between the factory calibration performed by EKO Instruments in 2016 and the IZO Langley-Plot calibration (2019) is shown in Figure 7. **These results indicate that the stability of the EKO MS-711 in the range 300-1100 nm during a 3 years period, between the manufacturer lamp calibration and the Langley calibrations at IZO, is remarkable...***"*

**Circumsolar radiation**

Circumsolar radiation contribution to the "true" measured direct irradiance is linked with AOD and also with aerosol types (phase functions). Higher AODs and forward scattering aerosols would introduce higher circumsolar correction factors. As in this work it is mentioned that a mixed (OPAC) based aerosol type is used, have you tested the actual correction and the effect on the AOD retrievals on a day with very high AOD and forward scattered aerosol type (e.g. dust aerosols) ?

*Authors: Yes. We have tested/validated the circumsolar radiation correction for dust and for different AOD intervals in the AOD range that we have been able to measure at IZO (up to 0.2). The results are shown in Figures 9 and 10 of the manuscript. Validations for data corrected by CSR are shown in blue*

**Line 41 GAW-PFR showing lower values**

*Authors: Done*

**Table 1 : cosine response : is that applicable to the DNI spectral measurements ?**

*Authors: No. It is not applicable to DNI measurements. The specifications given in Table 1 correspond to EKO MS-711 spectroradiometer measuring the global solar spectral radiation.*

**Lines 108-110: is this for direct or global irradiance?**

*Authors: It is for global irradiance*

**Lines 212: 0.09"**

*Authors: Done*

**Congratulations for a very interesting work.**

*Authors: Thank you very much for your comment.*